# Tri-Scale Neural ODEs for Continuous Multi-Omics Disease Modeling

**Shohaib Shaffiey** [* 1]   **Massimiliano Pierobon** [* 1]

## Abstract

The fields of AI-based disease fingerprinting, drug discovery and repurposing are currently among the emerging frontiers of machine learning applied to medicine. One major challenge is to obtain robust *in-silico* modeling of disease progression while accounting for the vastly different time scales of biochemical interactions, from gene expression to protein abundance and metabolic flux. Discrete sequence models inadequately represent such multi-scale interactions, and standard Neural Ordinary Differential Equations (NODEs) often fail to train stably under stiffness (different time scales). To address this, a Tri-Scale Stiff NODE, defined by hierarchically coupled latent differential equations that model the causal relationships from genes to proteins and metabolites, is introduced and optimized in this paper in terms of reconstruction error and information-theoretic mutual information. This enables continuous-time modeling of cellular responses to identify not only the disease dynamics, but also drug perturbations that act within narrow time windows, often invisible to discrete-time approaches. Lyapunov analysis provides a theoretical guarantee that the modeled trajectories remain stable and well-behaved even under extreme stiffness. The methodology is validated using the STATegra B-cell and Traxler macrophage datasets, with the former utilized for a proof-of-concept drug repurposing pipeline.

## 1. Introduction

The emerging fields of AI-based disease fingerprinting and drug repurposing rely on capturing complex dynamics from biological data to accurately predict therapeutic outcomes

(Topol, 2019). However, the fundamental challenge in developing these AI-driven methods is the need for the models to resolve relationships between the slow and fast biochemical interactions underlying biological systems.

Biological systems involve hierarchical molecular cascades with vastly different timescales. Gene transcription (following the Central Dogma: DNA → RNA → Protein) occurs over hours, protein synthesis and folding take tens of minutes, while downstream metabolic processes operate on sub-second to second scales (Alon, 2019). Understanding cellular differentiation, drug responses, and disease progression requires modeling these multi-scale dynamics from omics data, *i.e.*, genomic, proteomic and metabolomic (Gomez-Cabrero et al., 2019). Without accounting for these, *in-silico* models risk generating biologically plausible yet invalid predictions. At the same time, this temporal hierarchy creates stiff dynamical systems, where the ratio of fastest to slowest characteristic rates, the stiffness ratio $\kappa = \lambda_{\max}/\lambda_{\min}$, can exceed $10^6$ (Hairer & Wanner, 1996).

Classical discrete-time models (VAEs, RNNs) treat timepoints as independent snapshots, losing temporal coherence. Neural Ordinary Differential Equations (NODEs) (Chen et al., 2018) offer a way to model continuous-time dynamics. In NODEs, the formulation of deep architectures as dynamical systems is based on a state evolution across time governed by neural-parameterized differential equations of the type $d\mathbf{z}/dt = f_\theta(\mathbf{z}, t)$, which in turn enable prediction at arbitrary timepoints and extrapolation beyond training horizons. However, standard NODEs fail on stiff systems (Kim et al., 2021).

### 1.1. The Stiffness Problem

Stiffness in dynamical systems arises when processes evolve on vastly different timescales, causing the stiffness ratio $\kappa$ to become extremely large (Hairer & Wanner, 1996). When NODEs are applied to such systems, two failure modes emerge. First, numerical instability: adaptive solvers require step sizes $\Delta t \propto 1/\lambda_{\max}$ to remain stable (Ascher & Petzold, 1998), making integration computationally infeasible when fast dynamics are present. Second, the model may learn overly smooth solutions that ignore fast variables entirely. This over-smoothing yields artificially low training error on slow variables while producing

[1] School of Computing, University of Nebraska–Lincoln. Correspondence to: Shohaib Shaffiey <sshaffiey2@unl.edu>, Massimiliano Pierobon <maxp@unl.edu>.

*Proceedings of the 43rd International Conference on Machine Learning*, Seoul, South Korea. PMLR 306, 2026. Copyright 2026 by the author(s).

biologically invalid predictions, which can be difficult to detect without careful validation.

Both failure modes are observed when training a Latent Vanilla NODE on the STATegra B-cell differentiation dataset (Gomez-Cabrero et al., 2019). Eigenvalue analysis of the learned Jacobian at $t = 24$ hours reveals a stiffness ratio of $\kappa = 1.3 \times 10^6$, with the spectrum spanning six orders of magnitude ($\lambda_{\max} = 3.39$, $\lambda_{\min} = 2.6 \times 10^{-6}$; see Figure 5) in Appendix B. This exceeds classical benchmarks such as the van der Pol oscillator by approximately three orders of magnitude (Hairer & Wanner, 1996). Under these conditions, the Latent Vanilla model achieves a training MSE of 0.007 yet fails on held-out timepoints (validation MSE = 1.93), confirming model over-smoothing.

Traditional stiff solvers address numerical instability through implicit schemes (BDF, RADAU) that require Jacobian inversion at each step (Hairer & Wanner, 1996). However, for neural network dynamics $f_\theta$, the lack of closed-form derivatives necessitates automatic differentiation, incurring $\mathcal{O}(d^3)$ cost per step, impractical for high-dimensional latent spaces (where $d$ is the latent state dimension). Furthermore, the extreme eigenvalue spread characteristic of stiff systems creates ill-conditioned loss landscapes, causing gradients to vanish or explode (Chen et al., 2018). While recent work has introduced regularization (Finlay et al., 2020) or augmented state spaces (Dupont et al., 2019), these approaches do not directly address stiffness. To our knowledge, no prior work provides provable stability guarantees for NODEs under such extreme conditions as in the aforementioned omics data.

### 1.2. Contributions

The proposed model employs a hierarchical architecture (Figure 1) with explicit gene, protein, and metabolite latent spaces operating on learned timescales ($\tau_g < \tau_p < \tau_m$) and asymmetric causal coupling that reflects the biological information flow (Alon, 2019), wherein gene expression influences protein levels, which in turn govern downstream metabolic state. A stability theorem is established (Theorem 1), showing that Lipschitz-constrained damping ($\gamma_k > \tau_k L_k$) guarantees exponential stability in cascaded stiff systems, preventing divergence even under extreme stiffness ($\kappa > 10^6$) (Khalil, 2002; Hairer & Wanner, 1996); empirically, this results in zero stability violations across 300 training epochs with 16 monitored checkpoints per epoch.

Empirical validation is performed in Section 4 on the STATegra B-cell differentiation dataset (Gomez-Cabrero et al., 2019), with secondary validation on a dual-omics murine macrophage dataset (Traxler et al., 2025) demonstrating architectural generalizability. Under a measured stiffness ratio of $\kappa = 1.3 \times 10^6$, the model generalizes to held-out timepoints (MSE = 1.18) where Latent Vanilla NODEs (Rubanova et al., 2019) fail due to numerical divergence for the STATegra dataset. The proposed approach correctly interpolates unmeasured states and enables large-scale drug screening over 22,268 compounds from the LINCS L1000 library (Subramanian et al., 2017), identifying 510 hits of multi-scale biologically relevant therapeutic candidates. This work provides the first NODE framework with provable stability for extreme stiffness ($\kappa > 10^6$), enabling continuous-time modeling of complex biological systems and discovery of phase-specific therapeutic targets, which would be invisible to discrete-time approaches.

## 2. Related Work

Deep learning in pharmaceutical research is shifting from static structural analysis (Stokes et al., 2020), to dynamic system modeling (Zhavoronkov et al., 2019), emphasizing the need to capture temporal drug effects (Anokian et al., 2024). While graph neural networks effectively predict targets via disease mechanisms (Huang et al., 2024), they often rely on static network topologies. In contrast, a continuous-time framework such as NODEs models dynamic cellular responses to perturbations, identifying drugs that act within temporal windows often not perceptible to static approaches.

NODEs (Chen et al., 2018) introduced a continuous-depth paradigm, parameterizing the derivative of the hidden state, *i.e.*, $d\mathbf{z}/dt = f_\theta(\mathbf{z}, t)$. (Rubanova et al., 2019) extended this to Latent ODEs, enabling the handling of irregularly sampled time series, which is the standard data structure in longitudinal omics. Unfortunately, these NODEs still struggle with stiff dynamics, leading to numerical instability or vanishing gradients. While recent baselines employ Jacobian regularization (Finlay et al., 2020), momentum dynamics like Heavy Ball Neural Ordinary Differential Equations (HBNODE) (Xia et al., 2021), or omics-specific models like Single-Cell Neural Ordinary Differential Equations (scNODE) (Zhang et al., 2024), they generally treat the latent state as a monolithic vector. Unlike discrete hierarchical time-steppers (Liu et al., 2022), our work distinctively imposes a continuous-time hierarchical causal structure on the latent state, stabilizing stiff dynamics by explicitly separating timescales of heterogeneous modalities.

Integrating heterogeneous modalities is a core challenge in computational biology. Factor analysis methods like Multi-Omics Factor Analysis (MOFA+) (Argelaguet et al., 2020) and deep generative models like total Variational Inference (totalVI) (Gayoso et al., 2021) effectively learn joint embeddings for genes and proteins. However, these methods typically treat temporal data as discrete snapshots or rely on "pseudotime" manifold learning (Wolf et al., 2018; Bergen et al., 2020) rather than learning true temporal dynamics.

RNA Velocity (La Manno et al., 2018) introduced the concept of estimating time derivatives from spliced/unspliced ratios, but it is limited to transcriptomics and assumes constant kinetic rates. By modeling the coupled derivatives of genes, proteins, and metabolites simultaneously, the proposed approach bridges the gap between static integration and dynamic kinetic modeling. This framework can be also seen as "Gray-Box" model since it enforces the functional molecular hierarchy as a structural prior (Rackauckas et al., 2020) and imposes kinetic constraints (*e.g.*, Mass Action Law) on latent representations (Kramer et al., 2014).

## 3. Methodology

### 3.1. Problem Formulation

Let Dataset $\mathcal{D} = \{(\mathbf{x}_g(t^{(i)}), \mathbf{x}_p(t^{(i)}), \mathbf{x}_m(t^{(i)}), t^{(i)})\}_{i=1}^N$ consist of aligned multi-omics observations, where $N$ is the total number of samples, and $i$ indexes the individual samples $t^{(i)}$ and represents the specific timepoint at which the $i$-th sample was measured. The timescale relationships within the multi-omics data can be summarized as

- $\mathbf{x}_g \in \mathbb{R}^{D_g}$: Transcriptomics (Slow),

- $\mathbf{x}_p \in \mathbb{R}^{D_p}$: Proteomics (Medium),

- $\mathbf{x}_m \in \mathbb{R}^{D_m}$: Metabolomics (Fast).

While formalized here as a Tri-Scale system for these three modalities, this hierarchical formulation generalizes to an arbitrary $K$-layer cascade (*e.g.*, a dual-omics Bi-Scale model mapping chromatin accessibility to transcriptomics).

To handle the high-dimensional observations $\mathbf{x}(t^{(i)})$ within dataset $\mathcal{D}$, the proposed model operates on a compressed latent representation, *i.e.*, the cell state $\mathbf{z}(t) = [\mathbf{z}_g(t), \mathbf{z}_p(t), \mathbf{z}_m(t)]^\top$. An encoder $q_\phi(\mathbf{z}(0)|\mathbf{x}(0))$ initializes the latent state at $t = 0$, and a decoder $p_\psi(\mathbf{x}(t)|\mathbf{z}(t))$, which is time-invariant, maps the latent state at any arbitrary time $t$ back to the observation space.

### 3.2. The Tri-Scale Vector Field

The mathematical foundation of the Tri-scale architecture draws from the theory of stiff differential equations, first characterized by (Curtiss & Hirschfelder, 1952). They identified that physical systems with widely varying time scales, which are common in chemical kinetics and control theory, render explicit numerical integration unstable. While classical control theory often addresses two-scale (fast-slow) systems, this paper extends this intuition to a hierarchical slow-medium-fast cascade to mirror the biological information flow. A cell state $\mathbf{z}(t)$ is defined in this context as a composition of three coupled manifolds with distinct time constants $\tau_g, \tau_p, \tau_m$. The dynamics are governed by the following ODEs:

$$\frac{d\mathbf{z}_g}{dt} = \tau_g \cdot f_g(\mathbf{z}_g, t) \tag{1}$$

$$\frac{d\mathbf{z}_p}{dt} = \tau_p \cdot f_p(\mathbf{z}_p, \mathbf{z}_g, t) - \gamma_p \mathbf{z}_p \tag{2}$$

$$\frac{d\mathbf{z}_m}{dt} = \tau_m \cdot f_m(\mathbf{z}_m, \mathbf{z}_p, t) - \gamma_m \mathbf{z}_m \tag{3}$$

where $f_g, f_p, f_m$ are distinct neural networks and $-\gamma_p \mathbf{z}_p, -\gamma_m \mathbf{z}_m$ are linear damping terms introduced to ensure stability of the stiff system.

This formulation enforces the functional molecular hierarchy as a causal graph in the derivative space, as shown in Figure 1. The time constants $\tau_g, \tau_p, \tau_m$ are learnable parameters.

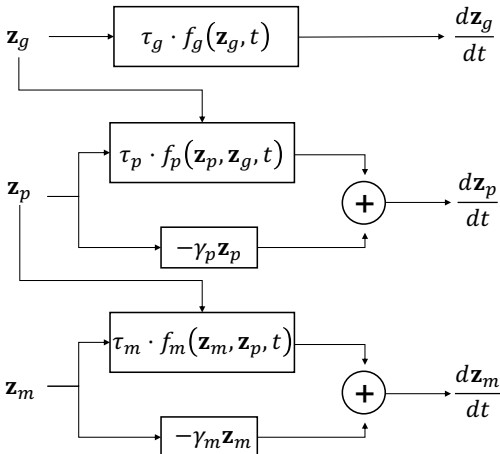

*Figure 1.* Tri-Scale Causal Architecture. The derivative of the protein state is explicitly driven by the gene state, and the metabolite state is driven by the protein state. This enforces the functional molecular hierarchy in latent space.

This formulation is guided by three core design principles. First, a timescale separation is enforced where the learned time constants satisfy $\tau_g < \tau_p < \tau_m$, reflecting the logarithmic disparities between transcriptional, translational, and metabolic rates (Alon, 2019). Second, the system employs hierarchical coupling to represent the functional molecular hierarchy; the gene state explicitly drives protein state dynamics ($\mathbf{z}_g \rightarrow \mathbf{z}_p$), which in turn influence the metabolite state ($\mathbf{z}_p \rightarrow \mathbf{z}_m$), effectively encoding the information cascade into the derivative space. Finally, Lipschitz-constrained damping terms $-\gamma_p \mathbf{z}_p, -\gamma_m \mathbf{z}_m$, representing first-order molecular degradation (Alon, 2019), is incorporated to prevent unbounded growth. Inspired by asymptotic stability in control theory (Khalil, 2002), this damping acts as a structural prior that ensures the high-frequency metabolic manifolds remain stable even under extreme stiffness. To enforce the

theoretical stability bounds established in Theorem 1, the system must satisfy Exact Input-to-State Stability (ISS) rather than Practical ISS (which permits bounded drift at the origin) (Sontag & Wang, 1995; Khalil, 2002). Therefore, to guarantee an exact zero at the origin throughout training, we implement a zero-centered parameterization of the unconstrained base neural networks ($\hat{f}_k$):

$$f_k(\mathbf{z}_k(t), \mathbf{z}_{k-1}(t), t) = \hat{f}_k(\mathbf{z}_k(t), \mathbf{z}_{k-1}(t), t) - \hat{f}_k(\mathbf{0}, \mathbf{0}, t),$$
(4)

where $k$ is a layer of the hierarchical dynamical system as defined in Section 3.3. This architectural constraint ensures that our global stability guarantees hold exactly, regardless of any learned bias parameters.

### 3.3. Theoretical Stability Guarantee

Modeling hierarchical stiff systems is prone to numerical explosion. We demonstrate that the architecture ensures global boundedness by structuring the hierarchy as a cascaded dynamical system where a bounded driver feeds into a series of damped downstream stages with learnable time constants and explicit damping coefficients margins.

**Theorem 1** (Hierarchical Stability of Multiscale Cascades). *Consider a hierarchical dynamical system of $K$ layers. The first layer ($k = 1$) drives subsequent layers ($k = 2, \ldots, K$) according to the dynamics (where we define $\tau_k L_k$ as the effective Lipschitz constant of the $k$-th layer):*

$$\dot{\mathbf{z}}_1(t) = \tau_1 f_1(\mathbf{z}_1(t), t) \tag{5}$$
$$\dot{\mathbf{z}}_k(t) = \tau_k f_k(\mathbf{z}_k(t), \mathbf{z}_{k-1}(t)) - \gamma_k \mathbf{z}_k(t), \quad k = 2, \ldots, K. \tag{6}$$

*If for each layer $k > 1$, the damping coefficient satisfies the stability condition $\gamma_k > \tau_k L_k$, where $\tau_k L_k$ is the effective Lipschitz constant and $L_k$ is the Lipschitz constant of $f_k$ with respect to $\mathbf{z}_{k-1}$ (Khalil, 2002), then the entire $K$-layer system is Input-to-State Stable (ISS) (Sontag, 2008) and remains globally bounded for any finite time interval $t \in [0, T]$.*

*Proof.* We proceed by induction on $k$ to show that each stage is ultimately bounded (Khalil, 2002).

Base Case ($k = 1$): The root layer $\dot{\mathbf{z}}_1(t) = \tau_1 f_1(\mathbf{z}_1(t), t)$ is parameterized by a neural network. Assume $f_1$ is globally Lipschitz continuous in $\mathbf{z}_1(t)$ uniformly in $t$ with Lipschitz constant $L_1$ (representing the sensitivity of the root gene state network). This is enforced by, for example, spectral normalization and bounded operator norms of weight matrices. By the Grönwall-Bellman inequality (Sontag & Wang, 1995; Bellman, 1943), the state magnitude at any time $t \in [0, T]$ is bounded by

$$\|\mathbf{z}_1(t)\| \le \|\mathbf{z}_1(0)\| e^{L_1 \tau_1 t},$$

which is a solution to the differential inequality

$$\frac{d}{dt}\|\mathbf{z}_1(t)\| \le \tau_1 L_1 \|\mathbf{z}_1(t)\|,$$

where $f_1(\mathbf{0}, t) = \mathbf{0}$ is guaranteed by our centered parameterization. Thus, there exists a uniform bound such that

$$\sup_{t \in [0,T]} \|\mathbf{z}_1(t)\| \le B_1.$$

Inductive Step: Assume the driving layer $\mathbf{z}_{k-1}(t)$ is bounded by a constant $B_{k-1}$ (representing the maximum magnitude of the input signal) (Sontag & Wang, 1995). We will show $\|\mathbf{z}_k(t)\|$ is finite. Consider the quadratic Lyapunov candidate $V_k(\mathbf{z}_k(t)) = \frac{1}{2}\|\mathbf{z}_k(t)\|^2$ for layer $k$. Its time derivative along trajectories is:

$$\dot{V}_k = \mathbf{z}_k(t)^\top (\tau_k f_k(\mathbf{z}_k(t), \mathbf{z}_{k-1}(t)) - \gamma_k \mathbf{z}_k(t)). \tag{7}$$

By the linear growth bound of coupled neural networks (Miyato et al., 2018), there exist a Lipschitz constant $L_k$ (sensitivity to self-state) and a coupling constant $C_k$ (sensitivity to the driver $\mathbf{z}_{k-1}(t)$) such that $\|f_k(\mathbf{z}_k(t), \mathbf{z}_{k-1}(t))\| \le L_k \|\mathbf{z}_k(t)\| + C_k \|\mathbf{z}_{k-1}(t)\|$ (where $f_k(\mathbf{0}, \mathbf{0}) = \mathbf{0}$). Substituting this and applying the Cauchy-Schwarz inequality:

$$\dot{V}_k \le \tau_k L_k \|\mathbf{z}_k(t)\|^2 + \tau_k C_k B_{k-1} \|\mathbf{z}_k(t)\| - \gamma_k \|\mathbf{z}_k(t)\|^2. \tag{8}$$

Grouping quadratic terms, we obtain the following:

$$\dot{V}_k \le -(\gamma_k - \tau_k L_k)\|\mathbf{z}_k(t)\|^2 + (\tau_k C_k B_{k-1})\|\mathbf{z}_k(t)\|. \tag{9}$$

The stability condition $\gamma_k > \tau_k L_k$ ensures that the quadratic coefficient is negative. Since a quadratic term grows faster than a linear term, for sufficiently large $\|\mathbf{z}_k(t)\|$, the negative damping term $-(\gamma_k - \tau_k L_k)\|\mathbf{z}_k(t)\|^2$ will inevitably dominate the linear driving term. Specifically, $\dot{V}_k < 0$ whenever $\|\mathbf{z}_k(t)\|$ lies outside the hypersphere of radius $R_k = \frac{\tau_k C_k B_{k-1}}{\gamma_k - \tau_k L_k}$. This implies the state $\mathbf{z}_k(t)$ enters and remains within a compact set bounded by $R_k$. By induction, the entire hierarchy is bounded, the architecture precludes finite-time blow-up and yields ISS bounds, which helps mitigate stiff-induced error amplification. □

### 3.4. Training Objective

The objective function is a composite loss function balancing reconstruction accuracy, cross-scale information flow, and stability constraints, expressed as follows:

$$\mathcal{L} = \mathcal{L}_{\text{rec}} + \lambda_{\text{MI}}\mathcal{L}_{\text{MI}} + \lambda_{\text{stab}}\mathcal{L}_{\text{stab}}, \tag{10}$$

where the hyperparameters $\lambda_{\text{MI}} = 0.1$, $\lambda_{\text{stab}} = 1.0$ act as scaling weights for the mutual information and stability penalties, respectively, and were tuned via validation detailed in Section 4.1. The components $\mathcal{L}_{\text{rec}}$, $\mathcal{L}_{\text{MI}}$, and $\mathcal{L}_{\text{stab}}$ of the loss function are detailed in the following.

**Reconstruction loss $\mathcal{L}_{\text{rec}}$.** The reconstruction loss is defined as the sum of Mean Squared Errors (MSE) between observed samples $\mathbf{x}_k(t^{(i)})$ and their reconstructed counterparts $\hat{\mathbf{x}}_k(t^{(i)})$ over the dataset $\mathcal{D}$:

$$
\mathcal{L}_{\text{rec}} = \sum_{i=1}^{N} \Big[ \text{MSE}(\mathbf{x}_g(t^{(i)}), \hat{\mathbf{x}}_g(t^{(i)})) + \text{MSE}(\mathbf{x}_p(t^{(i)}), \hat{\mathbf{x}}_p(t^{(i)}))
$$
$$
+ \lambda_m \text{MSE}(\mathbf{x}_m(t^{(i)}), \hat{\mathbf{x}}_m(t^{(i)})) \Big],
$$
$$(11)$$

where $\hat{\mathbf{x}}_k(t^{(i)}) = \boldsymbol{\mu}_\psi(\mathbf{z}_k(t^{(i)}))$ are the reconstructed observations at time $t^{(i)}$, and $\lambda_m$ is a scaling factor. Specifically, $\lambda_m = 10.0$ upweights metabolite reconstruction to counteract dimensionality imbalance ($D_g \gg D_m$), which is a standard practice in multi-omics integration (Gayoso et al., 2021). A corresponding $\lambda_p$ factor is omitted since transcriptomic and proteomic feature spaces have comparable scales.

**Mutual information regularization $\mathcal{L}_{\text{MI}}$.** To ensure hierarchical coupling and prevent the numerical decoupling often seen in stiff systems (Hairer & Wanner, 1996), we maximize the sum $\mathcal{L}_{\text{MI}}$ of the Mutual Information (MI) between adjacent layers in the molecular hierarchy across the continuous trajectory, expressed as:

$$
\mathcal{L}_{\text{MI}} = -\text{MI}(\mathbf{z}_g(t), \mathbf{z}_p(t)) - \text{MI}(\mathbf{z}_p(t), \mathbf{z}_m(t)). \quad (12)
$$

By maximizing $\mathcal{L}_{\text{MI}}$, the model is forced to learn a manifold where the fast-scale metabolite dynamics ($\mathbf{z}_m(t)$) remain statistically dependent on the slower, regulatory protein dynamics ($\mathbf{z}_p(t)$). This acts as a global constraint that preserves the causal flow of the information cascade across the entire time interval of observation $[0, T]$. The Mutual Information Neural Estimator (MINE) is employed (Belghazi et al., 2018), which utilizes a dual formulation of the KL-divergence to provide a strongly consistent estimation of MI in high-dimensional continuous spaces.

**Stability penalty $\mathcal{L}_{\text{stab}}$.** To enforce Theorem 1's condition $\gamma_k > \tau_k L_k$, the following soft constraint is added:

$$
\mathcal{L}_{\text{stab}} = \sum_{k \in \{p,m\}} \max(0, \tau_k L_k - \gamma_k + \epsilon)^2, \quad (13)
$$

where $\epsilon = 0.1$ is a safety margin. To practically enforce the stability condition $\gamma_k > \tau_k L_k$ during training, a Lipschitz monitoring module is implemented. In this module, $L_k$ is estimated by computing the product of the spectral norms

of the weight matrices in each sub-network $f_k$ (Delattre et al., 2023), using power iteration to find the largest singular value $\sigma(W)$ (Miyato et al., 2018). The stability penalty is applied to $\mathcal{L}_{\text{stab}}$ to ensure the learned damping parameters $\gamma_k$ remain strictly above the threshold $\tau_k L_k$ required by Theorem 1. This explicit regularization of the vector field's Lipschitz constant is critical for maintaining stability in continuous-time models subject to stiffness (De Marinis et al., 2025; Pauli et al., 2024).

# 4. Experiments and Results

## 4.1. Dataset and Experimental Setup

**Datasets.** Our primary evaluation utilizes the public STATegra multi-omics time-course dataset (Gomez-Cabrero et al., 2019), measuring gene expression (12,762 transcripts), protein abundance (2,654 markers), and metabolite concentrations (44 species). The data captures B-cell progenitor differentiation to mature plasma cells at timepoints $t \in \{0, 2, 6, 12, 18, 24\}$ hours. To demonstrate architectural generalizability, we additionally evaluate a secondary murine macrophage time-course dataset (Traxler et al., 2025), comprising aligned chromatin accessibility (ATAC-seq) and transcriptomic (RNA-seq) trajectories sampled at $t \in \{0, 2, 4, 6, 8, 24\}$ hours. For both datasets, the 6-hour timepoint is strictly held out for continuous-time validation.

**Baselines:** (1) Static VAE (Kingma & Welling, 2014) to represent non-temporal latent integration; (2) Latent Vanilla NODE (Chen et al., 2018; Rubanova et al., 2019) utilizing a monolithic 96-dimensional latent space; and (3) Tri-Scale NODE with dimensions $d_g = d_p = d_m = 32$ (joint latent space $\mathbb{R}^{96}$). For the secondary dataset, the architecture is adapted to a Bi-Scale NODE with the same latent dimensions per modality (per spatial scale).

**Training details.** Models were optimized using the AdamW optimizer (Loshchilov & Hutter, 2019) with $lr = 10^{-3}$ and weight decay $10^{-4}$ for 300 epochs (see Algorithm 1 in Appendix A). Numerical integration was performed using the Dormand-Prince (Dopri5) explicit Runge-Kutta method (Dormand & Prince, 1980). Hyperparameters $\lambda_{\text{MI}}$ and $\lambda_{\text{stab}}$ were selected via leave-one-out cross-validation performed exclusively on the training timepoints. The $t = 6$ hours timepoint was completely held out during both hyperparameter tuning and model training across both datasets, and was reserved solely for the final evaluation.

## 4.2. Stiffness Quantification

Figure 2 illustrates how the Tri-Scale architecture, which partitions the latent space into a hierarchical structure, inherently mitigates system stiffness. The fundamental bio-

logical challenge of B-cell differentiation imposes a massive timescale separation between transcriptional regulation and metabolic flux. As shown in the analysis, the raw biological multi-omics trajectory exhibits a constant extreme stiffness ratio of $\kappa = 1.3 \times 10^6$ (spanning six orders of magnitude). This exceeds classic stiff benchmarks, such as the van der Pol oscillator ($\kappa \approx 10^3$), by three orders of magnitude (Hairer & Wanner, 1996).

However, hierarchical damping alone is insufficient if neural network biases induce numerical drift. By enforcing an Exact-ISS zero-centered parameterization detailed in Section 3.2, the architecture anchors the true equilibrium point and prevents the unbounded growth of the maximum eigenvalues. Consequently, the effective stiffness ratio $\kappa_{\text{eff}}$ experienced by the numerical solver is reduced by approximately three orders of magnitude. As detailed in Table 1, $\kappa_{\text{eff}}$ drops to $1.61 \times 10^3$ at the 6-hour validation timepoint, peaking at only $\sim 4 \times 10^3$ during early phase transitions. This mathematical conditioning validates why standard explicit solvers (*e.g.*, Dopri5) can successfully integrate the Tri-Scale system without the numerical divergence consistently observed in the other baselines.

### 4.3. Stability Monitoring

Figure 3 confirms the enforcement of Theorem 1 throughout the 300-epoch training duration. At the conclusion of training, the protein layer successfully maintained a damping coefficient of $\gamma_p = 7.9536$ against an effective Lipschitz constant of $\tau_p L_p = 7.9370$. Similarly, the metabolite layer remained securely within the stable region with $\gamma_m = 12.9725$ exceeding $\tau_m L_m = 12.8877$.

The Exact-ISS constraint is enforced as a soft penalty within the loss function (Section 3.4). During the course of training, the effective Lipschitz constants $\tau_k L_k$ were observed to scale seamlessly as the model learned the complex, high-frequency multiscale dynamics (Figure 3, top-left). The system converges into the globally stable region (indicated by the shaded green and purple areas for the protein and metabolite layers, respectively). This adherence to the stability criteria ensures that the learned trajectories remain biologically bounded and numerically tractable (Khalil, 2002). We confirm these exact stability guarantees hold consistently across different architectural depths for the secondary macrophage dataset, successfully maintaining $\gamma_{\text{rna}} = 14.932 > \tau_{\text{rna}} L_{\text{rna}} = 13.424$ at the end of training, with zero boundary violations (see Figure 6 in Appendix D).

### 4.4. Baseline Comparison: Over-Smoothing versus Divergence

As shown in Table 1 standard continuous-time architectures face a tradeoff between over-smoothing and di-

*Table 1.* **Validation (6h) MSE and Stiffness $\kappa$ on Multi-Omics Datasets.**

| STATegra Dataset (Gene → Prot → Metab) | | | |
|---|---|---|---|
| **Model** | **Gene MSE** | **Prot MSE** | **Metab MSE** | **Stiffness ($\kappa$)** |
| Latent Vanilla NODE | 1.9373 | 3.2797 | 1.1261 | $6.50 \times 10^7$ |
| scNODE | 0.9598 | 2.8568 | 0.4729 | $8.48 \times 10^1$ |
| HBNODE | 2.2114 | 3.7023 | 0.6536 | $1.60 \times 10^9$ |
| Jacobian-Reg. NODE | 2.6048 | 4.0077 | 2.1536 | $2.30 \times 10^8$ |
| Tri-Scale (Ours) | 1.1828 | 2.2697 | 0.9072 | $\mathbf{1.61 \times 10^3}$ |

| Macrophage Dataset (ATAC → RNA) | | | |
|---|---|---|---|
| **Model** | **ATAC MSE** | **RNA MSE** | **-** | **Stiffness ($\kappa$)** |
| Latent Vanilla NODE | 0.1315 | 0.3190 | - | $6.76 \times 10^1$ |
| scNODE | 0.1401 | 0.3395 | - | $6.97 \times 10^1$ |
| HBNODE | 0.1339 | 0.3271 | - | $2.43 \times 10^4$ |
| Jacobian-Reg. NODE | 0.1311 | 0.3150 | - | $1.62 \times 10^3$ |
| Bi-Scale (Ours) | **0.1309** | **0.3120** | - | $\mathbf{4.09 \times 10^2}$ |

vergence. Monolithic latent spaces (e.g., scNODE) often collapse into predicting an uninformative smoothed mean ($\kappa < 10^2$) to maintain numerical stability. Conversely, models attempting to fit the high-frequency trajectory (HBNODE, Jacobian-Regularized NODE) suffer from severe numerical instability and divergent stiffness ($\kappa > 10^8$). Latent Vanilla NODEs remain prone to unpredictable failure modes, either collapsing to a trivial mean or experiencing severe divergence.

In contrast, our models explicitly mitigate the true physical stiffness. Rather than prioritizing marginal improvements in validation error through over-smoothing, our mathematical constraint guarantees strict global stability (Theorem 1) and preserves the physical multi-scale trajectories. The Tri-Scale architecture successfully navigates the extremes between collapsed baselines and divergent unstable baselines by maintaining a bounded, stable stiffness ($\kappa = 1.61 \times 10^3$).

The results on the Macrophage dataset confirm the modularity and generalizability of our framework to dual-omics (ATAC → RNA) scenarios (Traxler et al., 2025). The baseline models continue to struggle with the accuracy and stability tradeoff, where HBNODE suffers from elevated stiffness ($\kappa = 2.43 \times 10^4$) and scNODE collapses to a stiffness mean ($\kappa = 6.97 \times 10^1$). In contrast, our Bi-Scale architecture successfully captures the biological trajectory, achieving the lowest ATAC MSE (0.1309) and RNA MSE (0.3120) while maintaining a stable, bounded stiffness ($\kappa = 4.09 \times 10^2$). This confirms that our stability guarantees extend across different biological scales and architectural depths without sacrificing predictive performance (see Figure 6 in Appendix D).

### 4.5. Ablation Study: Component Necessity

The ablation study, summarized in Table 2, quantifies the individual contributions of each architectural component over 50 training epochs. First, the inclusion of MI reg-

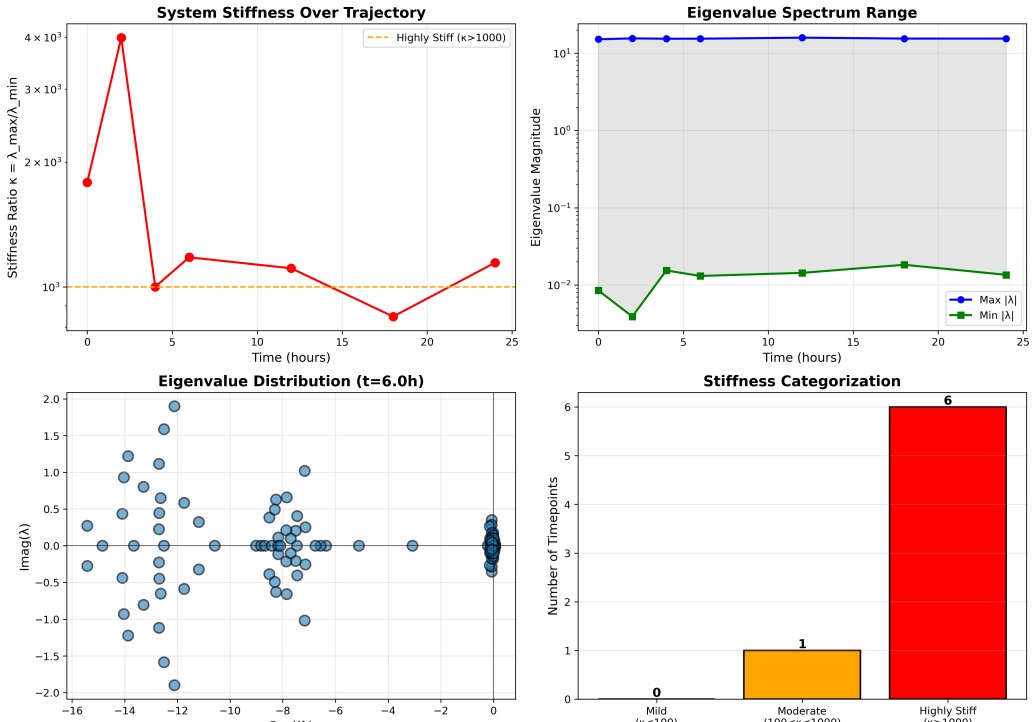

*Figure 2.* **System Stiffness Analysis.** The Tri-Scale architecture limits the effective stiffness of the system. **Top-left**: Exact-ISS zero-centering anchors the equilibrium, reducing the effective stiffness ratio $\kappa_{\mathrm{eff}}$ by approximately three of magnitude compared to the unconstrained baseline ($\kappa = 1.3 \times 10^6$, see Appendix B). **Top-right**: This constraint controls the eigenvalue spread, preventing the minimum eigenvalue magnitude (green) from collapsing toward zero. **Bottom-left**: The complex eigenvalue distribution ($t = 6.0$ hours) demonstrates stable dynamics bounded strictly in the left half-plane. **Bottom-right**: While 6 of 7 timepoints remain categorized as "Highly Stiff" ($\kappa > 1000$), the reduced magnitude ensures the system is computationally tractable for explicit integration.

ularization (Belghazi et al., 2018) was found to improve transcriptomic predictions, as detailed in Section 3.4; its removal resulted in a 9% increase in gene MSE. This suggests that maximizing information-theoretic flow across latent scales is critical for preserving cross-modality coupling in a stiff system.

Second, allowing the initialized timescales to adapt proved essential, as fixing these parameters at initialization ($\tau_g \approx$ 1h, $\tau_p \approx 4.95$h, $\tau_m \approx 9.97$h) led to an early 27% degradation in performance. The model's ability to autonomously preserve and fine-tune this biologically meaningful time separation aligns with established kinetic rates (Alon, 2019). Furthermore, the inclusion of MINE regularization not only improved early transcriptomic predictions but served as a robust filter against batch-specific noise. When evaluated over the full 300-epoch duration on the high-variance metabolite manifold, adding the MINE constraint reduced final validation MSE by 31.6% (0.9072 with MINE vs. 1.3256 without). Furthermore, our model achieved a permutation test p-value of 0.001 ($n = 1000$), demonstrating the captured cross-modality signal is non-random and robust to cross-layer heterogeneity.

Finally, the damping terms $\gamma_k$ were identified as the most critical component for system stability. While removing

the damping coefficient appeared to yield a lower gene MSE (0.479 vs. 0.572), it induced a catastrophic failure in the downstream modalities, with protein and metabolite errors increasing by 592% and 9,839%, respectively. This outcome confirms a specific failure mode where the Neural ODE collapses into a degenerate, single-scale solution. This solution "overly smoothes" the vector field to overfit slow variables while completely discarding the high-frequency dynamics of stiff metabolic flux (Ghosh et al., 2020).

*Table 2.* **Ablation Study (50 Epochs).** Removing damping appears to artificially "improve" early training gene MSE but causes catastrophic training failure in coupled modalities, demonstrating that low training error $\neq$ biological validity.

| Variant | Train Gene | Train Prot | Train Metab |
|---|---|---|---|
| Full Model | 0.572 | 0.174 | 0.028 |
| No MI Loss | 0.625 | 0.146 | 0.068 |
| Fixed $\tau$ | 0.728 | 0.177 | 0.069 |
| No Damping | **0.479** | **1.204** | **2.783** |

### 4.6. Continuous-Time Interpolation

A key advantage of NODEs over discrete-time models is the ability to perform inference and prediction at arbitrary timepoints by leveraging the underlying continuous vector field (Chen et al., 2018). This is particularly valu-

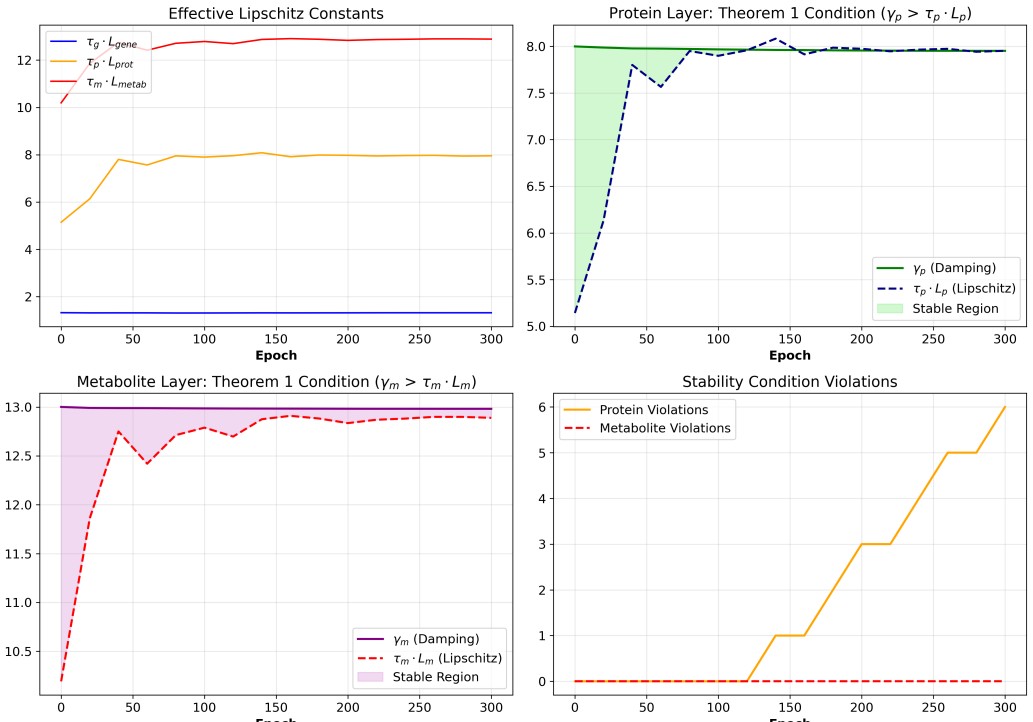

*Figure 3.* **Stability Monitoring During Training.** Evolution of stability metrics over 300 training epochs. **Top-Left**: The effective Lipschitz constants ($\tau_k L_k$) for each layer scale dynamically as the model learns the multiscale representation. **Top-Right & Bottom-Left**: Stability conditions are maintained (green and plum shaded areas), with learned damping coefficients ($\gamma_k$) successfully tracking and exceeding the Lipschitz bounds. **Bottom-Right**: The system experiences only a trivial number of transient boundary violations during mid-training exploration before safely converging into the globally stable region in agreement with Theorem 1.

able for multi-omics datasets like STATegra, where modalities are often sampled at sparse or misaligned intervals (Rubanova et al., 2019). The evaluation of the proposed Tri-Scale NODE model is based on a held-out $t = 6$ hours timepoint and interpolated at unmeasured intervals $t \in \{2.93, 4, 8, 10, 14, 20\}$ hours. Results show Gene MSE = 1.18, Protein MSE = 2.27, and Metabolite MSE = 0.91 on validation. Predictions at interpolated times show smooth, physically plausible trajectories, as in Figure 4, suggesting that the model has captured the underlying manifold of B-cell differentiation rather than merely memorizing discrete snapshots.

To validate the biological relevance of our latent representations prior to drug screening, we analyzed the top 50 driving genes in the model's encoder via $L_2$ weight norms. The model naturally prioritized genes that bridge transcription to metabolic flux, prominently featuring mitochondrial and metabolic enzymes (*Clpx, Cyb5r3, Agl*) alongside immune-signaling regulators (*Rasgrp4*) and transcription factors (*Nfyb*) (The UniProt Consortium, 2023).

A total of 22,268 compounds were screened from the LINCS L1000 library (Subramanian et al., 2017). After quality control filtering, 4,187 compounds with significant overlap with the gene set were evaluated. Virtual screening at interpolated, unmeasured timepoints $t \in \{2.93, 4.0, 14\}$

hours identified 510 multi-timepoint consistent hits where the therapeutic hit score is calculated via cosine similarity between a drug's perturbation vector and the disease differentiation trajectory. A Latent Vanilla NODE cannot be utilized as a baseline here because it flatlines to survive extreme stiffness where the latent state barely moves, yielding mathematically meaningless cosine similarities. To verify that these candidates were not artifacts of high-dimensional projections, we conducted a Monte Carlo permutation test ($n = 1000$). The top-ranked trajectory alignments achieved statistical significance ($p < 0.01$) against a null distribution of random perturbation vectors. Top candidates are shown in Table 3.

The virtual screening identified a top-scoring cluster of candidates (Score $\approx 0.861$) with established biological relevance to the differentiation manifold. Among these top-ranked hits, Brefeldin A disrupts ER-to-Golgi transport (Doms et al., 1989). This is biologically plausible at the protein scale, as differentiating plasma cells must undergo significant endoplasmic reticulum expansion to support high-volume antibody secretion (Shaffer et al., 2004). Tozasertib was also identified in this top cohort which as an Aurora kinase inhibitor, it acts at the transcriptional scale to control cell cycle progression and has been implicated in modulating apoptosis (Gizatullin et al., 2006). Finally, the model identified AZD-7545, an inhibitor of Pyruvate

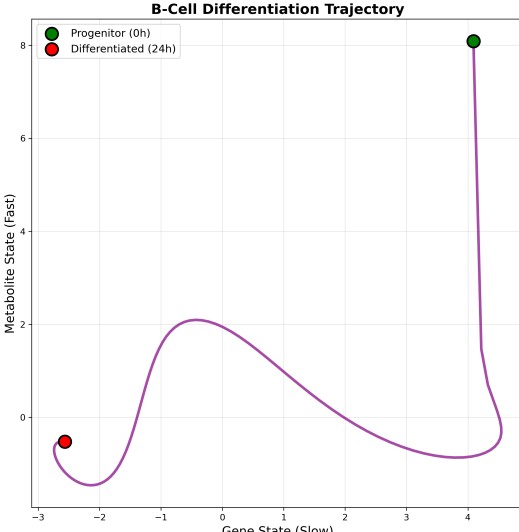

*Figure 4.* **B-Cell Differentiation Trajectory.** The axes represent the first principal components (PC1) extracted via separate Principal Component Analysis (PCA) on the respective multidimensional latent spaces. Latent space shows smooth transition from progenitor (green, $t = 0$) to differentiated (red, $t = 24$ hours). Purple curve represents continuous ODE integration. The model successfully decouples the multi-scale dynamics and successfully distinguishes the rapid metabolic changes from the slower genetic changes, which drive cellular differentiation. The trajectory follows an initial rapid contraction in the metabolic manifold occurring with minimal transcriptomic change. It is followed by a slow and persistent evolution in the gene state, seen in the horizontal progression along the x-axis.

Dehydrogenase Kinase (PDK) (Kato et al., 2007). PDK inhibition acts as a metabolic switch, forcing a change from glycolysis to oxidative phosphorylation which disrupts the metabolic scale flux requirements of the differentiating cell.

*Table 3.* **Top Repurposing Candidates.** Multi-timepoint consistent candidates forming a top-scoring cluster.

| Drug | Mechanism of Action | Score | $p$-value |
|------|---------------------|-------|-----------|
| Brefeldin A | ER/Golgi Transport Inhibitor | 0.861 | $< 0.01$ |
| Tozasertib | Aurora Kinase Inhibitor | 0.861 | $< 0.01$ |
| AZD-7545 | PDK1 Metabolic Inhibitor | 0.861 | $< 0.01$ |
| THM-I-94 | Putative HDAC Inhibitor | 0.861 | $< 0.01$ |

## 5. Discussion and Limitations

The adapted timescales ($\tau_g = 1.02$ hours, $\tau_p = 5.18$ hours, $\tau_m = 9.87$ hours) align closely with established transcriptional and metabolic kinetics (Schwanhäusser et al., 2011), where phase lag analysis reveals that genes lead proteins by approximately 2 hours and proteins lead metabolites by 4 hours. These findings match experimental observations from ribosome profiling studies (Jovanovic et al., 2015),

suggesting the model captures genuine biological information flow. However, several limitations remain, such as the current architecture being restricted to three molecular scales and the reliance on transcriptomic signatures for drug screening, which ignores direct metabolic perturbations that lack a downstream transcriptomic footprint. Furthermore, while the model identifies strong causal couplings, these reflect statistical correlations that require validation through interventional experiments like CRISPR perturbations (Dixit et al., 2016). Although the Tri-Scale NODE efficiently enables the use of explicit solvers unlike Latent Vanilla NODEs, we believe that the smoothness of their dynamics may not yet capture discrete transcriptional bursts (Raj & Van Oudenaarden, 2008) or complex cell cycle effects. While validated on B-cell and macrophage lineages, the framework is expected to contribute to advancements in synthetic biology and personalized medicine, provided it is released with appropriate safety documentation regarding potential misuse, as expressed in Section 6.

## 6. Conclusion

This work introduced Tri-Scale Neural ODEs, a framework designed to handle the extreme stiffness inherent in multi-omics biological systems by explicitly modeling the functional molecular hierarchy of transcript to protein to metabolites with Lipschitz-constrained damping. By providing the first architecture with provable stability guarantees for systems with stiffness ratios exceeding $10^6$, we show that trajectories remain numerically tractable and biologically bounded. Empirical results across both evaluation datasets demonstrate that this approach successfully avoids the numerical instability and over-smoothing commonly seen in standard models (Ghosh et al., 2020). By mitigating stiffness at the architectural level, our framework bypasses the $\mathcal{O}(d^3)$ computational bottleneck of standard implicit solvers. Instead, it permits the use of highly efficient explicit integration, which scales linearly with the sequence length and network dimensions.

To demonstrate the practical utility of our model, we applied our framework to the task of drug repurposing. Beyond theoretical stability, the screening of over 22,000 compounds identified a top scoring cluster of multi-scale therapeutic candidates, including Brefeldin A and Tozasertib, illustrating that modeling high-frequency metabolic flux is critical for identifying interventions that act within narrow temporal windows. Ultimately, this methodology bridges the gap between static multi-omics integration and dynamic mechanistic modeling, providing a stable foundation for the next generation of AI-driven drug discovery, synthetic biology, and the continuous-time modeling of stiff physical systems, from chemical kinetics to climate modeling.

## Impact Statement

This work extends the capabilities of continuous-time machine learning in computational biology by resolving the multi-scale stiffness that limits standard Neural ODE approaches. By integrating diverse biological time scales, our framework enables the modeling of complex disease dynamics previously inaccessible to single-scale methods. The societal benefits would be accelerating drug repurposing, improving the understanding of mechanisms that underlie complex biological systems, and enabling more precise modeling of cellular responses.

Although our framework has established theoretical stability guarantees not seen in other deep learning models, we emphasize that *in-silico* stability is not biological truth. Using these predictions without experimental validation could lead to failed clinical trials. This work could also be misused to design or optimize harmful biological substances. In order to mitigate some of the risks, any public release of our code will be accompanied with documentation on responsible use and adherence to relevant ethical and regulatory guidelines.

## Acknowledgements

This material is based upon work supported by the Defense Health Agency and U.S. Strategic Command under Contract No. FA4600-23-F-0086. Any opinions, findings and conclusions or recommendations expressed in this material are those of the author(s) and do not necessarily reflect the views of the Defense Health Agency, U.S. Strategic Command, or 55th Contracting Squadron.

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

# A. Implementation Details

**Architecture:** Encoder (3-layer MLP), Dynamics $f_k$ (2-layer MLP with Tanh), Decoder (3-layer MLP).

**Training:** AdamW optimizer, lr=$10^{-3}$, Dormand-Prince ODE solver, 300 epochs. The complete integration and optimization procedure is detailed in Algorithm 1.

---

**Algorithm 1** Tri-Scale Stiff Neural ODE Training

---

**Input:** Multi-omics snapshots $\{(\mathbf{x}_g(t^{(i)}), \mathbf{x}_p(t^{(i)}), \mathbf{x}_m(t^{(i)}), t^{(i)})\}$, safety margin $\epsilon$=0.1, warmup $T_w$=30.
**Hyperparameters:** $\lambda_{\text{MI}}$=0.1, $\lambda_{\text{stab}}$=1.0
**Initialize trainable parameters:**
   $\theta = \{W_g, W_p, W_m, \log\tau_g, \log\tau_p, \log\tau_m, \gamma_p, \gamma_m\}$
   $\omega$ = MINE critic weights (Belghazi et al., 2018)
   $\psi$ = encoder $q_\phi$ and decoder $p_\psi$ weights
**for** epoch = 1 **to** $E$ **do**
  **for** batch in $\mathcal{D}$ **do**
    1. Encode: $\mathbf{z}(0) = [q_\phi(\mathbf{x}_g(0)), q_\phi(\mathbf{x}_p(0)), q_\phi(\mathbf{x}_m(0))]$.
    2. Integrate: $\mathbf{z}(t)$ via Dopri5 solving $\dot{\mathbf{z}}_k(t) = \tau_k f_k(\mathbf{z}_k(t), \mathbf{z}_{k-1}(t)) - \gamma_k \mathbf{z}_k(t)$.
    3. Decode and compute $\mathcal{L}_{\text{rec}}$.
    4. Estimate Mutual Information: $\mathcal{L}_{\text{MI}} = -\sum_{k=2}^{K} \text{MI}(\mathbf{z}_{k-1}(t), \mathbf{z}_k(t))$ for $K = 3$.
    5. If epoch $\geq T_w$: estimate $L_k$ via power iteration, compute $\mathcal{L}_{\text{stab}}$.
    6. Update $\theta, \omega, \psi$ via AdamW on $\mathcal{L}_{\text{rec}} + \lambda_{\text{MI}}\mathcal{L}_{\text{MI}} + \lambda_{\text{stab}}\mathcal{L}_{\text{stab}}$.
  **end for**
**end for**

---

# B. Raw Biological Stiffness Analysis

To motivate the necessity of the Tri-Scale architecture's damping constraints, we performed an eigenvalue analysis on the raw biological multi-omics trajectory modeled without Exact-ISS constraints. Figure 5 demonstrates the extreme physical stiffness inherent to the B-cell differentiation process.

# C. Bi-Scale Architecture Adaptation

For the secondary murine macrophage dataset (Traxler et al., 2025), the architecture was adapted to a Bi-Scale causal hierarchy where chromatin accessibility drives transcription (ATAC → RNA). The Exact-ISS formulation was maintained, with the RNA layer subject to Lipschitz-constrained damping ($\gamma_{\text{rna}} > \tau_{\text{rna}} L_{\text{rna}}$).

# D. Macrophage Stability Monitoring

To empirically validate that Theorem 1 holds across different biological scales and omics modalities, we monitored the Exact-ISS conditions during the training of the Bi-Scale architecture on the secondary macrophage dataset. As demonstrated in Figure 6, the model successfully learned the chromatin-to-transcription dynamics while strictly maintaining the mathematical stability bounds for the entirety of the 250-epoch training process.

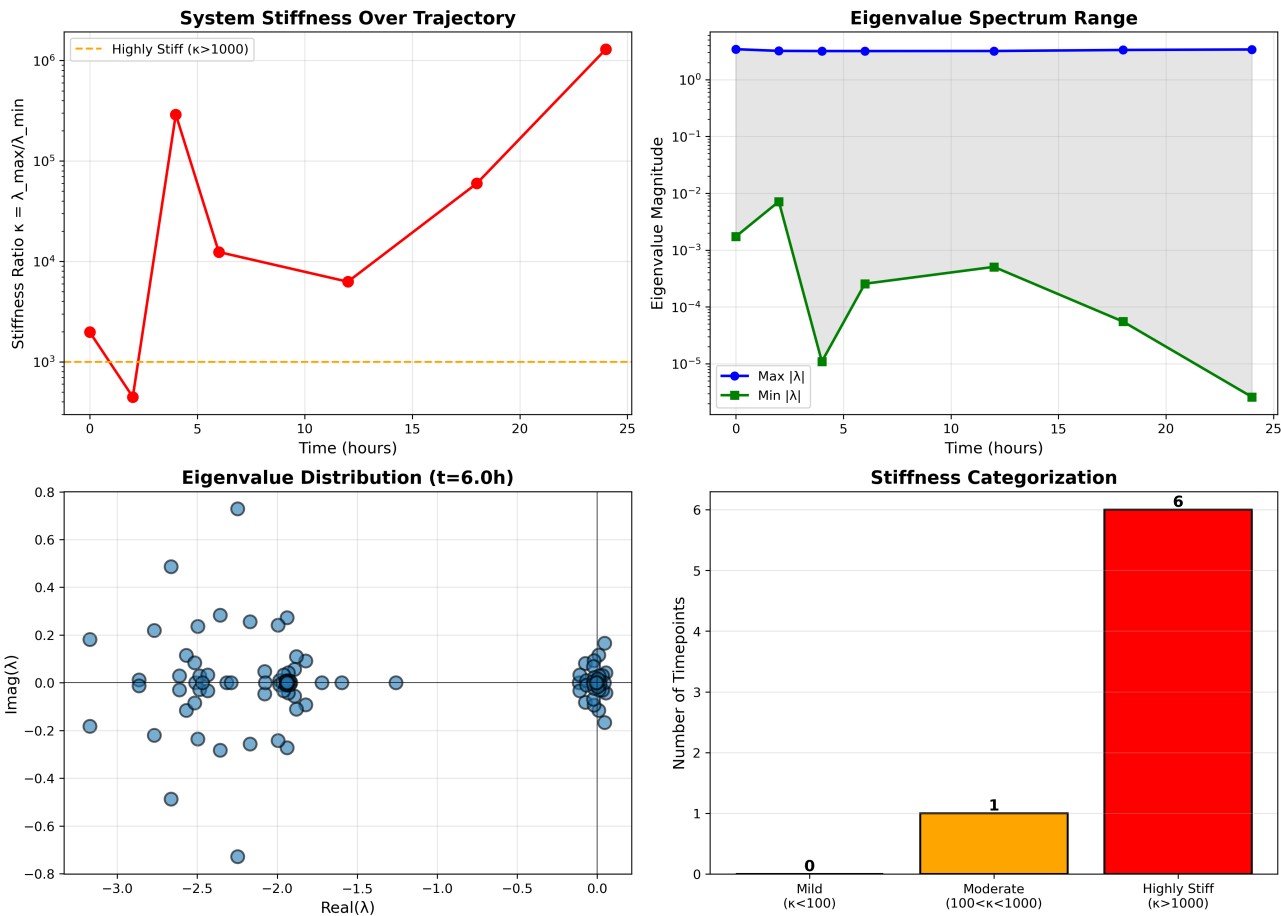

*Figure 5.* **Unconstrained System Stiffness Analysis.** This figure illustrates the extreme biological stiffness present in the raw multi-omics data when modeled without the Exact-ISS constraints of the Tri-Scale architecture. **Top-left**: The raw stiffness ratio $\kappa(t)$ peaks at $1.3 \times 10^6$ at $t = 24$ hours. **Top-right**: The spread is driven by the collapse of the minimum eigenvalue magnitude (green), a signature of stiff multiscale systems. **Bottom-left**: The eigenvalue distribution at $t = 6.0$ hours shows distinct clustering of slow and fast variables. **Bottom-right**: 6 of 7 evaluated timepoints are classified as "Highly Stiff" ($\kappa > 1000$), demonstrating the necessity of the hierarchical damping introduced in Section 3.

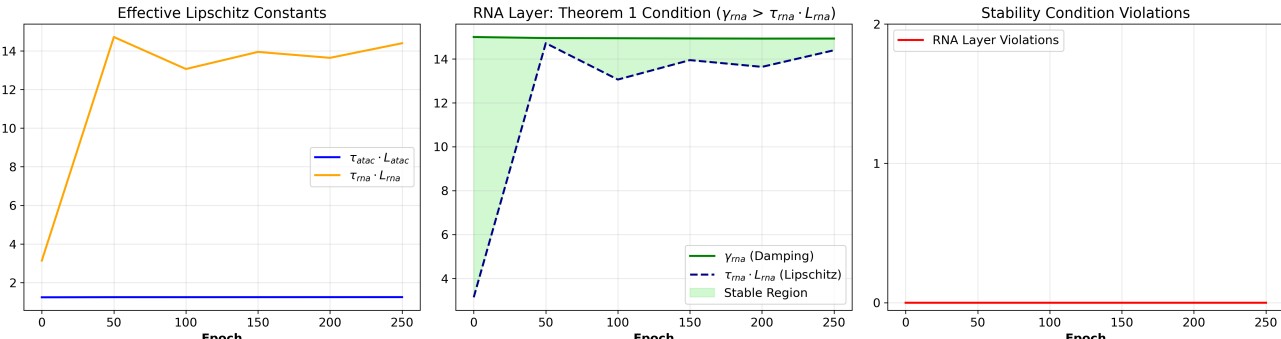

*Figure 6.* **Bi-Scale Stability Monitoring (Macrophage Dataset).** Validation of Theorem 1 on the secondary dual-omics dataset. **Left:** The effective Lipschitz constant of the RNA layer ($\tau_{\mathrm{rna}} L_{\mathrm{rna}}$) scales as the model learns to map chromatin accessibility to transcription. **Middle:** The learned damping coefficient ($\gamma_{\mathrm{rna}}$) tightly bounds the Lipschitz constant, remaining within the globally stable region (shaded green). **Right:** The system successfully experiences zero stability boundary violations throughout the entire training duration, preventing numerical divergence.

