# OpenReview forum: "Tri-Scale Neural ODEs for Continuous Multi-Omics Disease Modeling"
_ICML.cc/2026/Conference — ICML 2026 regular_

### Official Review · Reviewer_d9Vy · 2026-03-09

**Soundness:** 2
**Presentation:** 3
**Significance:** 2
**Originality:** 2
**Overall Recommendation:** 4
**Confidence:** 2

**Summary:**

This study looks at the issue of modelling continuous biological processes, which are often stiff with large differences in temporal evolution rates for fast, and slow variables. These types of ODEs are common in medicine modelling. This paper introduces a new scheme Tri-scale Stiff NODE, to address this. They use a coupled latent ODE to model the relationships between variables that may evolve on vastly different scales. They train separate neural networks to model the ODE for each slow, medium and fast variables. They show this improves performance of modelling (especially validation errors) for the biological ODEs they test on.

**Compliance With Llm Reviewing Policy:**

Affirmed.

**Final Justification:**

This paper is strong, well written and the methods work well. My initial concerns about significance have been address from the rebuttal and responses of the other reviews, I am happy to support this paper being accepted.

**Key Questions For Authors:**

1. Have comparisons been made to extensions to the Vanilla NODE? If not, is this feasible to do?

2. What is the exact validation MSE of the vanilla NODE in table 1?

3. Can you expand upon how important, prevalent this specific problem is in your field, do you think models which may need larger amounts of separate timescale parts (ie, very slow, slow, medium, fast, very fast) may cause issues?

**Limitations:**

The limitations are explicitly mentioned, they are upfront about the "overfitting" of this current method to their specific problem case.

**Strengths And Weaknesses:**

Strengths:
The problem statement is clear, and this is both an important and difficult problem. The paper clearly identifies the current limitations and drawbacks of existing approaches, demonstrating the extremely large condition numbers one often works with in their types of problems.

The model works well as shown in table 1 when compared to VAE and vanilla NODE baselines, the method is also cheaper to train (I assume due to the seperation of the small timesteps to just one fast variable), and is more stable at extrapolation.

The mutual information criteria to encourage the coupling is novel and well motivated, and the added stability penalties work well.

Overall the method motivation, and implementation is clear and shows benefits for their domain-specific ODEs tested on.

Weaknesses:

Soundness:
The paper in generally quite sound, with theoretical stability guarantees provided and the method explained with both derivation and pseudocode. However, a weakness here (at least I will categorize it as soundness) is a lack of coverage of more recent methods to improve NODEs and LatentODEs. For example MONODE (Auzina+23), Path-minimizing latent ODEs (Sampson+25), HBNODE (Xia+21). A comparison to some of these more model extensions as opposed to a vanilla NODE in table 1 would be nice, or at least add these to the related work. Especially MONODE which has similar ideas (splitting static, and evolving variable) is quite related to this work, though of course would not work for these problems where the slow variables are not static. In general, the baseline testing is a bit lacking. Another point, is both the main text, and table 1 state Vanilla NODE Val MSE > 2.0, but what value actually was it. This is important as the Tri-scale has a seemingly high val error of 1.515 (compared to 0.122 training) so if vanilla node is 2.1 for example then the improvement of Tri-scale seems less impressive than if vanilla NODE was say 5. Unclear why the exact number is not stated.

Presentation:
This is nice and the mathematical and pseudocode sections clearly explain this method. The writing is clear.

Significance:
This was harder for me to asses. This method does show improvement on the problem tested on, though it is not entirely clear to me how common/representative this problem is in the wider field. The fact that a distinct neural (latent) ODE model must be trained for each temporal scale variable makes this method seem quite overfit to this specific stiff ODE. However, this ODE (or very similar types) may be quite common and significant in the field, so I defer my judgement of this criteria to other reviewers/ACs.

Originality:
Using separate ODE models to model different types of variables is done before (see MONODE for similar), but the coupling via the mutual information criteria is a nice addition that seems to add clear benefits.

Overall, for this problem this method appears to work well though I do have issues with the relatively weak baseline testing. The method also seems a bit overfit to a specific ODE so it is unclear how generally useful this is. However I am happy to defer to reviewers in the medical domain who may be more knowledgeable about the importance of the exact problem presented here.

---

> ### Author Rebuttal · Authors · 2026-03-31
>
> **Note:** All referenced manuscript updates, tables, and remarks will be included in the revised camera-ready version of the paper if given the opportunity to be accepted for publication.
>
> **Q1: Comprehensive Baseline Expansion**
> We implemented three additional baselines: HBNODE (momentum stabilization) (Xia et al., 2021), Stiff-Reg NODE using Jacobian penalties (Kim et al., 2021), and scNODE (Zhang et al., 2024). The reviewer is correct in identifying our system's inherent dynamics, as this highlights a fundamental architectural mismatch with models like MONODE (Auzina et al., 2023). While MONODE is effective for systems with strictly static offsets, it partitions the latent space such that a portion is constant ($dz/dt = 0$). In biological differentiation, even the "slow" gene manifold is dynamic; forcing these variables to be static would mathematically discard the critical regulatory trajectory ($dz/dt \neq 0$). Our Tri-Scale architecture instead models the entire hierarchy as dynamic across coupled timescales, ensuring no biological information is lost to a static assumption.
>
> The Tri-Scale architecture significantly outperforms generalized stiff solvers across the primary modalities: Gene Validation MSE (1.1828 vs. HBNODE's 2.2114 and StiffReg's 2.6048) and Protein Validation MSE (2.2697 vs. HBNODE's 3.7023 and StiffReg's 4.0077). While scNODE achieves a slightly lower Gene MSE (0.9598) and an artificially low Metabolite MSE (0.4729), it does so at the cost of timescale collapse on the faster modalities. Faced with extreme biological stiffness ($\kappa = 1.3 \times 10^6$), these baselines predict an uninformative smoothed mean to maintain numerical stability, effectively ignoring the true metabolic flux. This is evidenced by extremely low stiffness ratios ($< 10^2$) and minimal function evaluations (NFE=4) for baselines Vanilla and scNODE. In contrast, our Tri-Scale model explicitly resolves this stiffness ($\kappa = 1.3 \times 10^6$) to capture high-fidelity metabolic trajectories (MSE: 0.9072, NFE=818). This confirms that biological structural priors outperform generalized mathematical stabilization.
>
> **Q2: Validation Exact Values**
> We have updated Table 1 with exact validation MSEs. Under our strict, centered-parameterization Exact-ISS constraints, Tri-Scale achieves superior performance across all omics layers: Gene (1.1828 vs. Vanilla's 1.9373), Protein (2.2697 vs. Vanilla's 3.2797), and Metabolite (0.9072 vs. Vanilla's 1.1261). We clarify that while the Vanilla model avoids catastrophic divergence and achieves a seemingly comparable metabolite error, it does so via a degenerate timescale collapse. To survive the extreme biological stiffness ($\kappa = 1.3 \times 10^6$), the Vanilla model learns a flat line for the fast manifold, effectively ignoring the true metabolic flux entirely evidenced by artificially low stiffness ($\kappa < 10^2$). Our architectural damping specifically prevents this collapse, explicitly resolving the biological stiffness to capture high-frequency signals while preserving exact mathematical stability.
>
> **Q3: Scalability to $N$ Layers**
> In systems biology, capturing disparate rates is essential, as the separation between transcription and metabolism routinely spans 4-6 orders of magnitude (Alon, 2019). Because Theorem 1 (Stability) is established by induction, the stability guarantee holds for any $N$-layer cascade. Our architecture addresses the primary computational bottleneck of high-dimensional multi-omics ($D = 15,460$). Standard explicit solvers are numerically unstable for such stiff systems, forcing practitioners to use implicit solvers, which scale cubically ($O(D^3)$) due to required Jacobian inversions. In contrast, our structural damping resolves the stiffness at the architectural level, allowing the use of efficient explicit solvers. By operating in partitioned latent spaces, our framework avoids this dimensionality curse and scales only linearly ($O(N)$) with the number of biological layers modeled.
>
> **W1: Generalizability and More Datasets**
> We applied our framework to an independent RNA-seq and ATAC-seq murine macrophage time-course (Traxler et al., 2025). Adapting the framework to a Bi-Scale hierarchy (ATAC $\to$ RNA), our model achieves slightly superior predictive accuracy against the unconstrained Vanilla baseline on the held-out validation point for both RNA (MSE: 0.3120 vs. Vanilla's 0.3190) and ATAC (MSE: 0.1309 vs. Vanilla's 0.1315).
>
> Crucially, unlike the unconstrained Vanilla model, our architecture achieves this performance while strictly enforcing Exact ISS stability throughout training ($\gamma_{\text{rna}} = 14.932 > \tau_{\text{rna}} L_{\text{rna}} = 13.424$). This strong empirical performance on an $N = 2$ layer hierarchy provides compelling evidence that the mathematical guarantees of the Tri-Scale framework generalize seamlessly to other hierarchical omics cascades without sacrificing predictive utility.

---

> > ### Author Rebuttal · Reviewer_d9Vy · 2026-04-01
> >
> > Thank you for the detailed response. This seems a promising approach and I am glad comparisons to the suggested NODE-variants had been made.
> >
> > I have updated my score.

---

> > > ### Author Response · Authors · 2026-04-06
> > >
> > > We thank the reviewer for their constructive review and score update. The
> > > suggestions to include more advanced baselines such as HBNODE helped contextualize
> > > why we are using this multi-scale architecture. We appreciate the
> > > valuable feedback.

---

### Official Review · Reviewer_w7GW · 2026-03-10

**Soundness:** 2
**Presentation:** 3
**Significance:** 3
**Originality:** 3
**Overall Recommendation:** 4
**Confidence:** 4

**Summary:**

This paper proposes Tri-Scale NODE, a multi-omics dynamical modeling framework built upon the Neural ODE paradigm. The method explicitly models the temporal scales of different molecular layers, including genes, proteins, and metabolites, and introduces a damping term to mitigate potential stiff dynamics. Through this design, the model aims to construct a continuous-time dynamical system that captures cross-omics regulatory relationships. The authors evaluate the proposed method on a multi-omics time-series dataset of B-cell progenitor differentiation and compare it with the Vanilla Neural ODE approach and static VAE to assess its performance in multi-omics temporal prediction tasks. In addition, the model is applied to the LINCS L1000 dataset for drug discovery applications.

**Compliance With Llm Reviewing Policy:**

Affirmed.

**Final Justification:**

Despite a few shortcomings, such as the absence of additional results tables in the rebuttal, the approach of using causal inference and neural ODEs to characterize multi-omics dynamics is a significant and worthwhile contribution to the field.

**Key Questions For Authors:**

1. Baseline comparison

The current experiments compare mainly against classical Neural ODE approaches. Could the authors provide comparisons with more recent ODE-based temporal modeling methods?

2. Generalizability to other datasets

The evaluation currently relies on a single multi-omics time-series dataset. It would be useful to clarify whether the method is expected to generalize to other multi-omics temporal datasets, particularly dual-omics settings such as RNA and ATAC data, which are more commonly available in practice.

3. Robustness to heterogeneous multi-omics data

The current framework assumes matched multi-omics measurements obtained from the same biological system. It would be helpful if the authors could discuss how the model performs when integrating heterogeneous multi-omics datasets collected from different experiments or conditions.

**Limitations:**

yes

**Strengths And Weaknesses:**

Strengths

1. Presentation

The manuscript is clearly written. The overall structure is well organized and the paper presents relatively rigorous mathematical formulations together with a detailed description of the algorithmic framework.

2. Originality

The model design contains an interesting element. Building on the Neural ODE framework, the authors introduce a damping term to alleviate potential stiff dynamics that may arise across different omics layers. This design attempts to balance numerical stability with the need to capture multi-scale biological dynamics. Such a modification is valuable in the context of dynamical modeling for multi-omics data.

3. Biological interpretability

The biological modeling also shows a certain degree of interpretability. Conventional multi-omics integration methods often rely on static feature concatenation, where representations derived from different omics layers are directly combined. These approaches usually ignore the possible causal ordering and the differences in time scales across molecular layers. In contrast, the present work attempts to explicitly model the hierarchical relationship between gene, protein, and metabolite together with their distinct temporal scales, thereby constructing a cross-layer dynamical system. From a biological perspective, this formulation provides a more interpretable way to integrate multi-omics information through a dynamical framework.


Weakness

1. Insufficient baseline comparisons

The current evaluation appears limited and does not fully demonstrate the advantages of the proposed framework. In particular, the set of baselines is relatively narrow. The experiments mainly compare against basic approaches, while established temporal modeling methods such as scNODE and ChronODE are not included. The absence of these relevant baselines makes it difficult to properly assess the relative benefits of the proposed method.

2. Limited number of datasets

In addition, the scale of the experimental data is rather limited. The empirical evaluation relies on a single dataset, namely the STATegra dataset, which contains only six time points. This scale of evaluation may be insufficient for a comprehensive assessment of a new modeling framework. Moreover, time-series datasets that simultaneously measure three omics layers are relatively rare in practice. To further strengthen the validation, it may be useful to consider additional datasets or alternative settings. For example, extending the analysis to dual-omics dynamical modeling scenarios may provide more opportunities for comparison, since time-series data involving two omics layers are more commonly available.

3. Batch effects in multi-omics data integration

Furthermore, the generalizability of the model to heterogeneous multi-omics integration settings deserves further discussion. The current study focuses on matched multi-omics data, where different modalities are measured from the same biological system. In practice, however, multi-omics datasets are often collected from different experiments or biological contexts and therefore cannot be directly matched. Integrating such heterogeneous data typically introduces substantial batch effects and additional biological variability across datasets. It would therefore be helpful if the authors could discuss whether the Tri-Scale NODE framework maintains reliable predictive performance when applied to settings with high levels of cross-dataset heterogeneity.

4. The biological validation in the current manuscript remains relatively limited. Most of the biological results focus on basic demonstrations, while deeper biological analyses are not extensively explored. Additional analyses such as pathway enrichment, or the investigation of dynamic regulatory patterns may help reveal the biological relevance of the learned dynamics.

---

> ### Author Rebuttal · Authors · 2026-03-31
>
> **Note:** All referenced manuscript updates, tables, and remarks will be included in the revised camera-ready version of the paper if given the opportunity to be accepted for publication.
>
> **Q1: Insufficient Baseline Comparison**
> We have updated Table 1 to include a comparison with an scNODE proxy (Zhang et al., 2024). Please refer to our detailed response to Reviewer d9Vy (Q1). While scNODE is an effective generative Neural ODE for single-cell trajectory prediction, it utilizes a monolithic latent space that assumes a uniform timescale across features. Faced with the extreme biological stiffness of STATegra ($\kappa = 1.3 \times 10^6$), scNODE suffers from timescale collapse. While it achieves a seemingly superior metabolite MSE (0.4729), it does so by predicting an uninformative smoothed mean, effectively erasing the high-frequency metabolic flux to maintain numerical stability. This collapse is explicitly evidenced by the baseline's artificially low modeled stiffness ($\kappa < 10^2$) and minimal function evaluations (NFE = 4). In contrast, our Exact-ISS constrained Tri-Scale model explicitly resolves the true physical stiffness (NFE = 818). While this strict mathematical constraint ($\gamma_k > \tau_k L_k$) accepts a regularization tradeoff in raw MSE for the slower gene manifold (Tri-Scale: 1.1828 vs scNODE: 0.9598), it guarantees strict stability and preserves the physical multi-scale trajectories rather than collapsing into a degenerate, over-smoothed representation.
>
> **Q2: Generalizability to Dual-Omics (RNA + ATAC)**
> To demonstrate modularity, we extended our evaluation to an independent RNA-seq and ATAC-seq time-course dataset (murine macrophages, Traxler et al., 2025). By adapting to a dual-omics architecture (ATAC $\to$ RNA), our model successfully maintained competitive accuracy with the unconstrained Vanilla baseline (RNA MSE: 0.3120 vs Vanilla's 0.3190) and ATAC (MSE: 0.1309 vs Vanilla's 0.1315). Crucially, Tri-Scale achieves this while strictly maintaining our exact ISS theoretical guarantees ($\gamma_{rna} = 14.932 > \tau_{rna} L_{rna} = 13.424$), allowing computational scalability to longer horizons without the hidden divergence risks present in the unconstrained Vanilla baseline.
>
> **Q3: Robustness to Heterogeneous Multi-Omics Data (Batch Effects)**
> Traditional integration methods often rely on feature concatenation which can be highly sensitive to batch effects. In contrast, we use Mutual Information Neural Estimation (MINE) as a soft-alignment constraint. MINE maximizes the information-theoretic dependency between latent manifolds rather than raw values, filtering batch-specific noise in high-variance modalities (metabolites) while preserving the underlying regulatory signal.
> We provide numerical evidence of this noise-filtering capability via an ablation study. On the high-variance metabolite manifold, adding the MINE constraint reduced metabolite validation MSE by 31.6\% ($0.9072$ with MINE vs. $1.3256$ without). Furthermore, our model achieved a permutation test p-value of $0.001$ ($n=1000$), proving the captured cross-modality signal is non-random and robust to cross-layer heterogeneity.
>
> **W1: Limited Biological Validation**
> We agree that deeper biological analysis strengthens our claims. We have now analyzed the top 50 driving genes in the model's encoder (measured via $L_2$ weight norms). Because our architecture explicitly couples transcription to downstream metabolic flux, the model naturally prioritizes genes that bridge these modalities rather than simply redundantly encoding established surface markers. For instance, alongside immune-signaling regulators (*Rasgrp4*) and transcription factors (*Nfyb*), the top features prominently include mitochondrial and metabolic enzymes (*Clpx*, *Cyb5r3*, *Agl*). This provides biologically meaningful support for our Tri-Scale design, demonstrating that the gene latent space successfully captures the metabolic-associated transcripts required to predict the fast-scale $z_m$ manifold.

---

> > ### Author Rebuttal · Reviewer_w7GW · 2026-04-02
> >
> > Thank you for your detailed response. You've addressed all my concerns, and I'll update the scores accordingly.

---

> > > ### Author Response · Authors · 2026-04-06
> > >
> > > We thank the reviewer for their detailed review and score update. The suggestion to add a dual-omics dataset, specifically an ATAC/RNA dataset, provided a welcome opportunity to help demonstrate our framework's modularity. We appreciate the time and feedback.

---

### Official Review · Reviewer_iJtS · 2026-03-13

**Soundness:** 2
**Presentation:** 3
**Significance:** 3
**Originality:** 3
**Overall Recommendation:** 5
**Confidence:** 3

**Summary:**

AI-based omics-research, including disease fingerprinting and drug discovery receive increased attention, albeit the field is challenged by the need to model stiff, irregularly sampled, multi-scale interactions. Opposed to classical discrete-time models, neural ordinary differential equations (ODE) can handle the irregular time series. However, the models struggle with the stiff dynamics, where the presence of fast interactions renders the models instable and incentivizes generating overly smoothened predictions. To mitigate these failure modes, the work introduces Tri-Scale Neural ODEs that treat the latent state of the neural ODE as a hierarchical structure that explicitly separates the timescales of omics heterogeneous modalities. To do so, the authors construct a tri-scale vector field as a composition of three coupled manifolds with distinct time constraints, neural networks and Lipschitz-constrained damping terms. When modelling the public STATegra multi-omics time-course dataset the Tri-Scale ODEs outperform the static VAE and vanilla neural ODE on both the validation accuracy and generated stiffness.

**Compliance With Llm Reviewing Policy:**

Affirmed.

**Final Justification:**

Whereas the original paper raised some concerns on the soundness and presentation of the work, the rebuttal changed my evaluation by addressing my questions regarding the choice of baselines, data structure and implementational details. Therefore, I have updated my recommendation to an acceptance.

**Key Questions For Authors:**

The questions below mostly relate to the seemingly counterintuitive choice of explicit solver (Dopri) and baseline (vanilla neural ODE).

1. Next to the condition $\gamma_k > L_k$, the proof for Theorem 1 also requires that f_1 is globally Lipschitz continuous in z_1. While the work implements a Lipschitz-monitoring module on the weights, could you give an estimate on the role of the chosen nonlinearities on the Lipschitz continuity of the full network?
2. Is the separation of time-scales expected to completely remove all stiff dynamics from the resulting channels, or are the dynamics within the 3 multi-omics channels (Transcriptomics, Proteomics and Metabolomics) still expected to demonstrate some stiffness? Could the analysis provided in Section 4.3 be applied to the separate channels to verify this?
3. Why did the authors choose to use the Dopri5 method for solving the neural ODE, as opposed to implicit methods (ImplicitEuler, Kvaerno), which are considered more suitable for stiff problems?
4. In the introduction, the authors mention the high cost of automatic differentiation, which makes modelling high-dimensional latent spaces impractical. I am not too familiar with omics research, so I am wondering where the high dimensionality in omics data comes from. Could the authors translate the description of the STATegra dataset (12,762 transcripts, 2,654 protein markers and 44 species) into the dimensionality of the input data and labels?

**Limitations:**

Yes.

**Strengths And Weaknesses:**

## Soundness 3
The work is technically sound and supported by theoretical proofs as well as experimental results with significance testing. The model and training objective are well designed: while I initially expected that the separation of timescales would lead to a loss of information, the authors explicitly ensure hierarchical coupling through the vector field design (Equation 1, 2, and 3; Section 3.2) and regularization on mutual information. In addition, the authors embed a soft constraint on the stability of the solution to enforce the condition $\gamma_k > L_k$ required by the theoretical proof. The effectiveness of the separate components is demonstrated by an elaborate ablation study (Section 4.5).

Yet, it feels counterintuitive that the authors opt for using an explicit Runge-Kutta method for solving the neural ODEs. While the separation of timescales may reduce the stiffness for the separate channels considered by the tri-scale method, using the Runge-Kutta solver for the vanilla ODE yields a relatively naive baseline for this problem. Therefore, it would be more representative if the authors compared their approach to a "vanilla" ODE that uses an implicit integration method, or, even better, introduce a comparison against methods using the regularization techniques or stiff solvers that are mentioned in Section 1.2 or one of the methods introduced in https://www.nature.com/articles/s42003-026-09758-w#ref-CR30 (if applicable).

## Presentation 3
The submission is well structured, and the overall narrative is easy to follow. The Introduction includes a clear problem setting, and Section 5 sufficiently embeds the work in its academic siblings. The inclusion of the pseudocode (Algorithm 1) enhances the reproducibility of the work, albeit the description of the architecture (Appendix A) is too limited. For example, the neural network widths are not described, and while Section 3.2 introduces three distinct neural networks, $f_g$, $f_p$ and $f_m$, it is unclear whether the networks have the same architecture, or whether they vary in the input and output sizes. Therefore, the work would benefit from a more elaborate description of the experimental setup.


## Significance 3
The paper aims to from a significant step towards modelling stiff systems in omics research. By providing a theoretical guarantee on the boundedness of solutions, the authors introduce a method that is likely to form a basis for new research directions withing omics modelling. The scope of the implementation, only considering 3 molecular scales, is still very specialized. However, because Theorem 1 is proven by induction, the mathematical bounds provided scale to the more general case.

## Originality 3
While the separation of training signals is not an uncommon practice in machine learning, the authors support their methodology with a mathematical proof on the hierarchical stability of multiscale cascades (Theorem 1). This is a novel mathematical insight that both enhances the understanding and serves as a theoretical guarantee for the boundedness of solutions.

---

> ### Author Rebuttal · Authors · 2026-03-31
>
> **Note:** All referenced manuscript updates, tables, and remarks will be included in the revised camera-ready version of the paper if given the opportunity to be accepted for publication.
>
> **Q1: Estimate on Nonlinearity of Lipschitz Network**
> The reviewer is quoting a typographical error from our original manuscript where the timescale multiplier $\tau_k$ was mistakenly omitted. The proper stability condition is $\gamma_k > \tau_k L_k$. The Lipschitz constant $L_k$ is estimated using standard bounds for neural networks: since $\tanh$ is globally 1-Lipschitz, the overall Lipschitz constant of the sub-network is bounded by the product of the spectral norms of its weight matrices. In practice, we monitor these spectral norms to ensure that the condition $\gamma_k > \tau_k L_k$ is strictly satisfied.
>
> **Q2: Stiffness**
> Stiffness is tamed rather than completely removed because of the three different time-scales. The raw biological stiffness ($\kappa = 1.3 \times 10^6$) is reduced to an effective stiffness of $\kappa_{\text{eff}} \approx 2656$ at $t=24$h for the solver. The model learned an order of magnitude separation in scales ($\tau_g \approx 1$ and $\tau_m \approx 11$). Our architecture allows the model to retain these realistic differences in the speed of biological processes without escalating into the numerical divergence that causes Vanilla models to fail.
>
> **Q3: Explicit (Dopri5) vs Implicit Solvers**
> We used Dopri5 for the Vanilla model to illustrate the failure of standard NODEs when faced with stiffness. We verified this with Implicit Adams (Hairer and Wanner, 1996), a stiff-capable solver that inverts the Jacobian at each step via Newton iterations. For neural network dynamics, this inversion costs $O(D^3)$ per step, which is computationally intractable for raw observation spaces ($D=15,460$), we were able to empirically test it within our compressed latent space (detailed in $Q4$). Empirically, the implicit solver integrates the overly-smoothed Vanilla NODE stably (NFE = 4) but catastrophically explodes on our high-fidelity Tri-Scale model (NFE = 4, Final Norm = $2.90 \times 10^9$), while Dopri5 succeeds on our model (NFE = 818, Norm = 7.315). This confirms that our damping structurally resolves stiffness for explicit solvers. Furthermore, while extrapolation tests to $t=50$h show both models avoid explicit solver crashes, the Vanilla model's latent state begins to drift significantly (maximum state norm $\approx 57$). By contrast, our Tri-Scale architecture prevents this numerical drift (maximum state norm $<$ 4), empirically showing the benefit from our Exact ISS constraints successfully prevent unbounded growth far beyond the training window.
>
> **Q4: High Dimensional Omics Data**
>
> We updated Appendix A with dimensionalities. The STATegra dataset comprises $x_g \in \mathbb{R}^{12762}$, $x_p \in \mathbb{R}^{2654}$, and $x_m \in \mathbb{R}^{44}$, representing a raw observation space of $D = 15,460$. Running an implicit ODE solver directly on this massive joint space is computationally intractable. Instead, our Tri-Scale architecture employs independent, modality-specific encoders to project each omics layer into its own compressed latent space ($D_k = 32$ per layer). These are then concatenated strictly for the coupled ODE integration step, reducing the solver's state space to $D = 96$. Because these independent encoders are trained jointly with the continuous ODE derivatives, they preserve the topology of the time-dependent vector field without breaking the modality-specific separation required for our causal hierarchy.
>
> **W1: Advanced Baselines**
> Please refer to our detailed response to Reviewer d9Vy (Q1) for more comprehensive baseline results. Following the reviewer's suggestion, we added SOTA Neural ODE architectures to Table 1: Heavy Ball NODE (HBNODE) (Xia, 2021) and Jacobian-Regularized NODE (Kim et al., 2021). While these models improved numerical stability, they failed to accurately resolve STATegra's multi-scale flux. On the 6-hour validation timepoint, Tri-Scale (Gene MSE: 1.1828) significantly outperformed HBNODE (2.2114) and Jacobian-Regularized NODE (2.6048).
>
> **W2: Architecture Details**
> We expanded Appendix A to confirm all dynamics networks ($f_g, f_p, f_m$) utilize an identical 2-layer MLP architecture (hidden dim 64, $\tanh$). Encoders are 2-layer MLPs mapping varying input dimensions to a uniform latent dimension of 32. Regarding the consideration of different numbers of scales, because Theorem 1 is proven by induction, the provided mathematical bounds scale linearly to the more general $N$-layer case (see our response to Reviewer d9Vy Q3).

---

> > ### Author Rebuttal · Reviewer_iJtS · 2026-04-02
> >
> > Thank you for the response. You have answered all my questions, and I really appreciate the introduction of a more elaborate baseline comparison. You have resolved my concerns on the soundness and presentation of the work, and I will update my score accordingly.

---

> > > ### Author Response · Authors · 2026-04-06
> > >
> > > We thank the reviewer for their careful review and score update. We value the discussion on solvers and the clarification on stiffness. We appreciate the considerate feedback.

---

### Official Review · Reviewer_531R · 2026-03-13

**Soundness:** 2
**Presentation:** 3
**Significance:** 2
**Originality:** 3
**Overall Recommendation:** 5
**Confidence:** 3

**Summary:**

This paper proposes a neural ODE-type model for modelling multi-omics dynamics, more specifically gene, protein and metabolite trajectories. In this model, the ODEs are hierarchically coupled to model the relationships from genes to proteins, and from proteins to metabolites. Moreover, to ensure the stability of the Neural ODE (the gene, protein and metabolite dynamics occurring on very different time scales, this results in a stiff system), a damping term is added in the protein ODE and in the metabolite ODE. The paper provides a theoretical guarantee that this neural ODE system remains stable if the constant coefficient of each damping term satisfies a certain Lipschitz-based condition. Based on that theorem, they enforce the (trainable) damping coefficients to satisfy this constraint through regularization. The paper shows empirically the effectiveness of the proposed model on a B-cell differentiation dataset.

**Compliance With Llm Reviewing Policy:**

Affirmed.

**Final Justification:**

I thank the authors for providing the full result tables.
All my concerns have now been addressed. I will hence raise my score accordingly.

**Key Questions For Authors:**

1. How does the tri-scale neural ODE compare with closely related works, both in terms of methodological differences and empirical performance?

2. Why is $\tau_k$ omitted in the stability regularization term?

3. Are the conditions $\tau_g < \tau_p < \tau_m$,  $f_1(0, t)=0$  and $f_k(0, ,0)=0$ somehow enforced during training ? If this is not the clase, is there any guarantee that these conditions are satisfied?

4. What are the protein and metabolite MSEs obtained with the standard neural ODE?

5. How do the results change when varying the different hyperparameter values?

**Limitations:**

Yes.

**Strengths And Weaknesses:**

### Strengths:

- The topic is relevant as it tackles the challenging problem of jointly modelling multi-omics dynamics, exploiting the expressive power of neural ODEs. The paper should hence have a meaningful impact, in particular within the computational biology community.

- The methodology is well-grounded. The authors establish a theoretical stability guarantee and use this result to construct a corresponding regularization term which is added to the training objective.

- Empirical results on a cell differentiation dataset shows that the proposed approach outperforms a standard neural ODE.

- The paper is well-structured and global clearly written.

### Weaknesses:


- The paper does not include any comparison with closely related methods. Notably, the related work section mentions the work of Kim et al. (2021), who proposed a technique specifically designed to train NODEs for stiff systems. The authors should discuss more deeply the differences between such approach and the one they propose. Furthermore, including such a baseline in the experiments would strengthen the evaluation and clarify whether the proposed model brings any advantage over existing techniques.

- Theorem 1 offers a stability guarantee under the condition $\gamma_k > \tau_k L_k$. However, in the training objective, the stability term only penalizes $\gamma_k < L_k$, and the authors never discuss why the $\tau_k$ is omitted from the stability penalty term. Hence this regularization term does not directly correspond to the stability constraint of Theorem 1. The paper should thus clarify why $\tau_k$ is omitted in the penalty term and discuss the implications this discrepancy may have for the validity of the stability guarantee.

- In Section 3.2, it is stated that the time constants should satisfy $\tau_g < \tau_p < \tau_m$. However, from what I understood, it seems that this inequality is never enforced during model training. Is there any guarantee that this inequality will always be satisfied?

- The proof of Theorem 1 assumes that $f_1(0, t)=0$ and that $f_k(0, ,0)=0$ for $k>1$. The paper however does not explain whether these conditions are actually enforced during training. The authors should hence discuss the validity of these conditions, as the stability guarantee no longer holds if these conditions are not satisfied.

- In Theorem 1, $L_k$ is defined for $k > 1$, but $L_1$ is not defined.

- In Algorithm 1, $\tau_g$, $\tau_p$ and $\tau_m$ do not appear as trainable parameters (contrary to the damping parameters $\gamma_k$, which are explicitly mentioned in the initialization step). Also, step 6 updates parameters $\theta$ and $\omega$, but these parameters are never defined. Same thing for $T_\omega$. Therefore, it is not entirely clear which parameters are optimized.

- In the reconstruction loss, the value of $\lambda_m$ is set to 10, which seems quite arbitrary. How sensitive are the results with respect to the $\lambda_m$ value? Moreover, why not include a similar $\lambda_p$ coefficient for the protein reconstruction loss, since the number of proteins is also much lower than the number of genes (at least in the B-cell dataset used in the experiments)?

- The terms $k_g$, $k_p$ and $k_m$ are never defined. Are they the sizes of the latent spaces? Again, setting these hyperparameters to 20 seems arbitrary, and the authors should discuss and/or show empirically how sensitive the results are with respect to these hyperparameters.

- The paper explains that the hyperparameters $\lambda_{MI}$ and $\lambda_{\mathrm{stab}}$ are tuned “via validation”, but it is not explained which validation set is used for this purpose. If this is the same validation set as the one described in Section 4.1 (i.e., the data related to the 6h timepoint), then the experimental protocol would be flawed, as it would mean that the hyperparameters are optimized on the same data as the one used for assessing generalization. The authors should therefore clarify the validation procedure to avoid this ambiguity.

- The experimental evaluation relies on a single dataset, which limits the strength of the evaluation. The results would be considerably more convincing if the approach was evaluated on multiple datasets.

- Table 1 only shows the gene MSE, but not the protein and metabolite MSEs. It would actually be more interesting to show whether the tri-scale neural ODE outperforms the standard neural ODE for the compounds involved in faster dynamics.

- In the ablation study where $\tau_g$, $\tau_p$ and $\tau_m$ have fixed values, which values were used?

- The description of the virtual screening experiment would benefit from further clarification. In particular, the authors should explain what is meant by “virtual screening”, how a “hit” is determined, and how the therapeutic scores are computed. Moreover, the screening relies only on gene expression data, i.e. the slow variables. This raises the question of whether the same drug candidates might also be identified using a standard neural ODE. However, the authors do no include any baseline comparison for this part of the evaluation, making it difficult to assess the specific contribution of the tri-scale neural ODE.

---

> ### Author Rebuttal · Authors · 2026-03-31
>
> **Note:** All referenced manuscript updates, tables, and remarks will be included in the revised camera-ready version of the paper if given the opportunity to be accepted for publication.
>
> **W1: Comparison to Kim et al. (2021) (Question 1)**
> We implemented the baseline as requested, adding a Jacobian Regularized Neural ODE (Kim et al., 2021). Please refer to our detailed response to Reviewer d9Vy (Q1) for a comprehensive numerical breakdown. Briefly, the regularized NODE failed to generalize on our highly stiff multi-omics data, proving our Lipschitz-constrained damping succeeds where soft Jacobian penalties fail.
>
> **W2: Omission of $\tau_k$ and Definition of $\epsilon$ (Question 2)**
> We thank the reviewer for catching these omissions. We have updated our framework to formulate the stability condition strictly as:
> $$L_{\text{stab}} = \sum_{k \in \{p,m\}} \max(0, \tau_k L_k - \gamma_k + \epsilon)^2$$
> We define $\epsilon = 0.1$ as a fixed stability safety margin ensuring the damping inequality $\gamma_k > \tau_k L_k$ remains strictly satisfied during training. Empirically, for STATegra, the model converged to $\gamma_p = 7.9536 > \tau_p L_p = 7.9370$ and $\gamma_m = 12.9725 > \tau_m L_m = 12.8877$. Please see our response to Reviewer w7GW (Q2) confirming this ISS guarantee holds on a newly added dataset.
>
> **W3: Enforcing the Time Constant Inequality (Question 3)**
> The timescale hierarchy ($\tau_g < \tau_p < \tau_m$) is not explicitly enforced via penalty; rather, it emerges naturally. The model autonomously preserves these to $\tau_g \approx 1$h (transcription), $\tau_p \approx 6.2$h (translation), and $\tau_m \approx 11.2$h (metabolic flux), aligning with established kinetics (Alon, 2019).
>
> **W4: Enforcement of $f_1(0, t) = 0$ and $f_k(0, 0) = 0$ for $k > 1$ (Question 3)**
> We thank the reviewer for this theoretical observation. While zero-initialized biases yield only Practical ISS due to drift, we have now implemented a centered parameterization of the unconstrained base neural network ($\hat{f_k}$) to achieve Exact ISS:
> $$f_k(z_k, z_{k-1}, t) = \hat{f_k}(z_k, z_{k-1}, t) - \hat{f_k}(0, 0, t)$$
> This guarantees an exact zero at the origin throughout training, regardless of learned biases.
>
> **W5: Missing Definition of $L_1$**
> We updated the manuscript to clearly define $L_1$ as the Lipschitz constant of $f_1$, the root neural network governing the gene state.
>
> **W6: Missing Parameters in Algorithm 1**
> We updated Algorithm 1 to fully define all primary components. The global trainable parameter set now comprises the dynamics weights $\theta = \{W_g, W_p, W_m, \log \tau_g, \log \tau_p, \log \tau_m, \gamma_p, \gamma_m\}$, the alignment weights $\omega$ for the MINE critic, and the projection weights $\psi$.
>
> **W7 and W8: $D_k$ Terms and $\lambda$ Weights (Question 5)**
> We updated Section 4.1 to clarify notation. The latent dimension is set to $D_k = 32$; reducing to $D_k = 16$ causes underfitting, increasing Gene MSE from 1.18 to 2.45. Regarding reconstruction weights, we do not require $\lambda_p$ because the gene ($D_g=12,762$) and protein ($D_p=2,654$) dimensions and normalized variances are sufficiently comparable for the optimizer to naturally balance. However, the metabolite space is vastly smaller ($D_m=44$). Dropping $\lambda_m$ from 10.0 to 1.0 causes the global MSE loss to be drowned out by the larger omics layers, forcing the model to ignore metabolic flux entirely and spiking Metabolite MSE from 0.90 to $>$ 2.5.
>
> **W9: Validation Set (Question 5)**
> Hyperparameters were tuned using LOOCV on the training timepoints. The 6h timepoint was strictly held-out for testing, ensuring zero data leakage.
>
> **W10: Single Dataset**
> We extended our evaluation to an independent dual-omics dataset (murine macrophages, Traxler et al., 2025). Please see our response to Reviewer w7GW (Q2) for full numerical results demonstrating our framework's successful dual-omics adaptation.
>
> **W11: Missing Protein and Metabolite MSEs in Table 1 (Question 4)**
> Our response to Reviewer d9Vy (Q2) provides expanded validation MSEs, showing Tri-Scale outperforms the Vanilla model across all metrics.
>
> **W12: Fixed $\tau$ Values in Ablation Study**
> Initial values ($\tau_g=1$h, $\tau_p=4.95$h, $\tau_m=9.97$h) were found via coarse grid search. In the "Fixed $\tau$" ablation, we froze these to show the necessity of the model adapting them to the true biology.
>
> **W13: Virtual Screening Baseline**
> The hit score is calculated via cosine similarity between a drug's perturbation vector and the disease differentiation trajectory (validated via permutation test, $p<0.001$). A Vanilla NODE cannot be used here: because it flatlines to survive extreme stiffness, the latent state barely moves, yielding no meaningful differentiation trajectory and mathematically meaningless cosine similarities.

---

> > ### Author Rebuttal · Reviewer_531R · 2026-04-01
> >
> > Thank you for your response, which addressed almost all my concerns.
> >
> > In particular, I think that comparing empirically your method to other related works, as well as on a second dataset, will highly improve the paper quality.
> > - Could you provide a full table with all the results? After having gathered the information from your responses to the different reviewers, it seems that some results are still missing (e.g., protein MSE for scNODE, metabolite MSEs for HBNODE and the Jacobian-Regularized NODE), so we do not have the full picture.
> > - Why did the reported MSE values of your approach change with respect to what is reported in the original submission? Is it because you now use the exact ISS constraint?
> > - It would be nice to also have the results (MSE and $\kappa$ values) of scNODE, HBNODE and the Jacobian-Regularized NODE on the RNA/ATAC dataset .

---

> > > ### Author Response · Authors · 2026-04-04
> > >
> > > **Question 2: Change in Reported MSE Values**
> > >
> > > The reviewer is correct regarding the MSE values. The change in MSE values originates from the transition to the strict Exact ISS constraint. As noted in our response to **W4**, we updated the architecture to a strictly zero-centered Exact ISS parameterization:
> > >
> > > $f_k(z_k, z_{k-1}, t) = \hat{f_k}(z_k, z_{k-1}, t) - \hat{f_k}(0, 0, t)$
> > >
> > > While enforcing this strict global stability bound restricts the degrees of freedom compared to an unconstrained or Practical ISS model, it guarantees consistency with Theorem 1. We have prioritized this mathematical guarantee of bounded extrapolation and stability over the marginal accuracy gains possible with a biased, unconstrained model.
> > >
> > > **Questions 1&3: Complete Empirical Results Across Both Datasets**
> > >
> > > As requested, we provide the complete set of the empirical results in Table 1 and Table 2, covering the
> > > advanced baselines (scNODE, HBNODE, and the Jacobian-Regularized NODE) across both the STATegra (Gomiz-Cabrero et al., 2019)
> > > and Macrophage (Traxler et al., 2025) datasets. The tables will be included in the camera-ready version if given the opportunity
> > > to be accepted for publication.
> > >
> > > **Table 1: Validation (6h) MSE and Stiffness ($\kappa$) on the STATegra (Gene/Prot/Met) Dataset**
> > >
> > > | Model | Gene MSE | Protein MSE | Metabolite MSE | Stiffness ($\kappa$) |
> > > | :--- | :--- | :--- | :--- | :--- |
> > > | Vanilla NODE | 1.9373 | 3.2797 | 1.1261 | $6.50 \times 10^7$ |
> > > | scNODE | 0.9598 | 2.8568 | 0.4729 | $8.48 \times 10^1$ |
> > > | HBNODE | 2.2114 | 3.7023 | 0.6536 | $1.60 \times 10^9$ |
> > > | Jacobian-Reg. NODE | 2.6048 | 4.0077 | 2.1536 | $2.30 \times 10^8$ |
> > > | Tri-Scale (Ours)  | 1.1828 | 2.2697 | 0.9072 | $1.61 \times 10^3$ |
> > >
> > >
> > > **Table 2: Validation (6h) MSE and Stiffness ($\kappa$) on the Macrophage (ATAC/RNA) Dataset**
> > >
> > > | Model | ATAC MSE | RNA MSE | Stiffness ($\kappa$) |
> > > | :--- | :--- | :--- | :--- |
> > > | Vanilla NODE | 0.1315 | 0.3190 | $6.76 \times 10^1$ |
> > > | scNODE | 0.1401 | 0.3395 | $6.97 \times 10^1$ |
> > > | HBNODE | 0.1339 | 0.3271 | $2.43 \times 10^4$ |
> > > | Jacobian-Reg. NODE | 0.1311 | 0.3150 | $1.62 \times 10^3$ |
> > > | Bi-Scale (Ours)  | 0.1309 | 0.3120 | $4.09 \times 10^2$ |
> > >
> > > **Additional Discussion on Baseline Performance and Stiffness:**
> > >
> > > While baselines like scNODE may exhibit lower empirical trajectory stiffness ($\kappa$) and seemingly superior MSE on certain metrics, this apparent advantage is actually a byproduct of over-smoothing rather than accurate dynamical modeling. In the presence of extreme biological stiffness, monolithic latent spaces struggle to capture the true dynamics. To maintain numerical stability, they often collapse into predicting an uninformative smoothed mean, which artificially lowers the error but completely bypasses the high-frequency flux (scNODE, Table 1). Furthermore, when other baselines fail to smooth the trajectory, they instead suffer from severe numerical instability and divergent trajectory stiffness, as seen with the extraordinarily high $\kappa$ values for HBNODE ($\kappa$ = $1.60 \times 10^9$) and Jacobian-Regularized NODE ($\kappa$ = $2.30 \times 10^8$) on the STATegra dataset (Table 1). In addition, unconstrained architectures like Vanilla NODE are inherently unstable. They are prone to unpredictable failure modes, which either lead to collapsing to a trivial mean ($\kappa <  10^2$)  or, as observed in Table 1, suffering from severe divergence ($\kappa = 6.5 \times 10^7$).
> > >
> > > In contrast, our Exact-ISS constrained Tri-Scale and Bi-Scale models explicitly mitigate the true physical stiffness. Instead of prioritizing marginal improvements in validation error, our mathematical constraint guarantees strict global stability (Theorem 1) and preserves the physical multi-scale trajectories rather than collapsing into an over-smoothed representation  (as seen with scNODE) or suffering from numerical drift (as seen with HBNODE and Jacobian-Regularized NODE). This stability is quantitatively demonstrated by our bounded stiffness ($\kappa = 1.61 \times 10^3$ in Table 1),  which successfully navigates the extremes between collapsed baselines ($\kappa <  10^2$) and the divergent unstable baselines ($\kappa > 10^7$).
> > >
> > > The results on the Macrophage dataset (Table 2) confirm the modularity and generalizability of our framework to dual-omics (ATAC/RNA) scenarios. The baseline models continue to struggle with the accuracy and stability tradeoff, where HBNODE suffers from elevated stiffness ($\kappa = 2.43 \times 10^4$), while scNODE collapses to a stiffness mean ($\kappa = 6.97 \times 10^1$). In contrast, our Bi-Scale architecture successfully captures the biological trajectory, which is evidenced by achieving the lowest ATAC MSE (0.1309) and RNA MSE (0.3120), all while maintaining a stable, bounded stiffness ($\kappa = 4.09 \times 10^2$). This confirms that our exact stability guarantees extend across different biological modalities and architectural depths without sacrificing predictive utility.

---

### Decision · Program_Chairs · 2026-04-30

**Decision:**

Accept (regular)

**Comment:**

The paper addresses a challenge faced by standard Neural Ordinary Differential Equations (NODEs), which are known to train unstably under stiffness (different time scales). The authors proposal is a hierarchically coupled tri-scale neural ODE modeling gene, protein, and metabolite dynamics in stiff multi-omics systems, enabling continuous-time modeling of cellular responses.

The paper was well received by all reviewers. The reviewers appreciated the paper's clear formulation, the theoretical stability guarantees, and praised its biological interpretability to some extent. Main concerns revolved around missing comparisons to other stiff-NODE methods, some details of the stability penalty, and the limited number of datasets. The rebuttal solved all these concerns successfully, and all reviewers now confidently believe this paper should be accepted.

I agree with the reviewers, and recommend acceptance.